

Environmental dynamics since the last glacial period in arid Central Asia:
evidence from grain size distribution and magnetic properties of loess
from the Ili Valley, western China
Yue Li[1,2], Yougui Song[1]*, Kathryn E. Fitzsimmons[3], Hong Chang[1], Rustam Orozbaev[4,5], Xinxin Li
[1,2]
1 State Key Laboratory of Loess and Quaternary Geology, Institute of Earth Environment, Chinese
Academy of Sciences, Xi'an, 710061, China
2 College of Earth Science, University of Chinese Academy of Sciences, Beijing, 100049, China
3 Research Group for Terrestrial Palaeoclimates, Max Planck Institute for Chemistry, Hahn-
Meitner-Weg 1, 55128 Mainz, Germany
4 Research Center for Ecology and Environment of Central Asia, Chinese Academy of Sciences,
Urumqi, 830011, China
5 Institute of Geology, National Academy of Sciences, Bishkek, 720040, Kyrgyzstan
* Corresponding author:
Dr. Yougui Song
E-mail: syg@ieecas.cn
State Key Laboratory of Loess and Quaternary Geology
Institute of Earth Environment, Chinese Academy of Sciences
No. 97 Yanxiang Road, Yanta, Xi'an 710061, China
Tel. 86-29 6233 6216
Fax. 86-29 6233 6216





**Abstract**

The extensive loess deposits of the Eurasian mid-latitudes provide important terrestrial records of Quaternary climatic change. As yet, however, loess records in Central Asia are poorly understood. Here we investigate the grain size and magnetic characteristics of loess from the Nilka (NLK) section in the Ili Basin of eastern Central Asia. Magnetic parameters indicate very weak pedogenesis compared with loess from other regions in Eurasia. The higher $\chi_{lf}$ values occur in primary loess, rather than in weak paleosols, and the variations in magnetic susceptibility (MS) value correlate closely with the proportions of the sand fraction. We attribute this result to high wind strength at the time of loess deposition. To explore the dust transport patterns further, we identified three grain size end members (EM1, mode size 47.5 μm; EM2, 33.6 μm; EM3, 18.9 μm) which represent distinct aerodynamic environments. EM1 and EM2 represent the grain-size fractions transported from proximal sources in short-term, near-surface suspension during dust outbreaks. EM3 appears to represent the continuous background dust fraction under non-dust storm processes. Of the three end members, EM1 is most likely the most sensitive recorder of wind strength. A lack of correlation between EM1 proportions and GISP $\delta^{18}O$ values at the millennial scale, combined with modern weather data, suggests that Arctic polar front predominates in the Ili Basin and the Kyrgyz Tian Shan piedmont during cold phases, which leads to the dust transport and accumulation of loess deposits, while the shift of mid-latitude westerlies towards the south and north controls the patterns of precipitation/moisture variations in this region. Comparison of EM1 proportions with Northern Hemisphere summer insolation clearly illustrate local insolation-based control on wind dynamics in the region, and humdity can also influence grain size of loess over MIS3 in particular. Although, the polar front dominated wind dynamics for loess deposition in the region, the Central Asian high mountains obstructed its migration further south. Our results may also support the significance of the mid-latitude westerlies in transmitting North Atlantic climate signals to East Asia.

Key words: Last glacial, Ili Basin, Central Asia, loess, magnetic susceptibility, grain size, paleoclimate

**1 Introduction**

Central Eurasia experiences extremely continental climatic conditions in large part due to its position far from oceans. Arid Central Asia is therefore a sensitive recorder of past climate change due to its location in the transition zone between the Asian monsoon (Dettman et al., 2001;Cheng et al., 2012), mid-latitude westerlies (Vandenberghe et al., 2006) and North Asian polar front (Machalett et al., 2008). The relative influence and intensity of these major climate subsystems have varied across the latitudinal and longitudinal range of Central Asia through time. Thus identification of the predominant climate regimes in a certain region is a crucial precondition for tracing paleoclimatic evolution.

One of the most promising potential palaeoenvironmental archives in the Central Asian region is its widespread, thick loess deposits. Loess is one of the most important archives of Quaternary climate change (Maher, 2016;Muhs, 2013). The semi-arid zone of Eurasia, between 45° and 30° N, hosts some of the thickest and most extensive loess deposits in the world. In Central Asia, the loess deposits cover the slopes of the Tian Shan mountains, from Xinjiang province of China and Kazakhstan, to Kyrgyzstan and Uzbekistan, to Tajikistan. While loess in Central Asia has



increasingly formed the focus of loess research (Dodonov et al., 2006;Feng et al., 2011;Li et al.,
2016c;Li et al., 2016b;Machalett et al., 2006;Smalley et al., 2006;Song et al., 2014;Song et al.,
2015;Song et al., 2012;Yang et al., 2006;Youn et al., 2014;Fitzsimmons et al., 2016), as yet the
forcing mechanisms and the climatic conditions responsible for loess-paleosol sequences formation
are ambiguous, and the paleoclimatic evolution recorded by these loess deposits in this region is not
systematically understood.
Evidence for temperature oscillations associated with the Greenland (*Dansgarrd/Oeschger (D-*
*O) events*) (Dansgaard et al., 1993) and cool phases associated with iceberg calving into the North
Atlantic (*Heinrich (H) events*) (Bond et al., 1992) have been found in loess deposits based on the
high-resolution grain-size variations ether in Chinese Loess Plateau (CLP) loess (Sun et al.,
2012;Porter and An, 1995) or in European loess (Antoine et al., 2009;Rousseau et al., 2007;Zeeden
et al., 2016). Climatic teleconnections, especially between the North Atlantic and East Asian
Monsoon regions, are likely to have been recorded within the Central Asian loess. As yet, however,
the region so far largely lacks data by which the role and contribution of the central parts of the
Eurasian continent, as an environmental bridge, can be elucidated.
The Ili Basin of Central Asia represents a region of thick loess deposits with high potential for
investigating palaeoenvironmental change for the region. The situation of the basin, surrounded to
the south and north by the Tian Shan mountain range and widening to the west (Fig. 1), provides a
conducive situation for loess accumulation which has resulted in the widespread and thick loess
deposits in this basin. In this paper we present new data on the physical properties of a 20.4 m thick
loess deposit at Nilka (NLK) in the eastern Ili Basin, focusing on grain size distributions and
magnetic properties in order to investigate the enhancement mechanisms of magnetic susceptibility
in NLK loess and elucidate environmental dynamics based on grain size data.
**2 Physical geography**
The Ili Basin (80° ~ 85° E and 42° 30′ ~ 44° 30′ N) straddles southeast Kazakhstan and
northwest China. It is an intermontane basin opening westward towards the semi-arid Kazakhstan
Gobi Desert which forms the transitional region between the steppe and full deserts of Central Asia.
The Northern and Southern Tian Shan form the northern and southern boundaries to the basin (Fig.
1a). The Ili River drains northwestward into terminal Lake Balkhash.
This region has a semi-arid, continental climate, with a strong precipitation gradient dependent
on altitude. The altitude of the basin floor is 500 ~ 780 m; the northern Tien Shan Range reaches
altitudes of > 4000 m a.s.l. and the southern Tien Shan mountains range between 3000 ~7000 m
a.s.l. towards the catchment divide. The mean annual precipitation (MAP) ranges between 200 mm
and 500 mm on the plains, and mean annual temperature (MAT) ranges from 2.6°C to 10.4°C (Li,
1991;Ye, 1999). The surface vegetation in this region is dominated by *Desert Steppe* and *Steppe* and
the zonal soils comprise *Sierozem*, *Castonozem* and *Chernozem*.
The Nilka (NLK) section (83.25°E, 43.76°N, 1253 m a.s.l) is situated on the second terrace of
the right bank of the Kashi River, a tributary of the Ili River. The site is located in the eastern Ili
Basin of far western China, adjoining the Northern Tian Shan to the north (Fig. 1b).

Fig. 1 The location of study area and the photo of Nilka (NLK) section.

**3 Materials and methods**
**3.1 Section and sampling**



The NLK loess section has a thickness of 20.4 m and overlies fluvial sands and gravels (Fig.1).
The profile has been exposed recently by local residents for making bricks, and recently formed the
focus of a geochronological study comparing luminescence with radiocarbon methods (Song et al.,
2015). According to the dating results of Song et al. (2015), the NLK loess started to accumulate
since ~ 70 ka B.P.. Stratigraphically and geochronologically, this is equivalent to the L1 loess unit
(known as Malan loess) and S0 paleosol unit (known as Holocene Heilu soil) in the Chinese Loess
Plateau, 2300 km to the east. Although largely homogeneous in appearance, two weak paleosols (at
5.04 − 7 m and 15.7 − 18 m depths) were identified in the section by field observations and
confirmed by our subsequent grain-size and magnetic susceptibility (MS) results. We therefore
divided the NLK stratigraphy into S0, L1L1, L1S1, L1L2, L1S2 and L1L3 units (Fig. 1c).
Following cleaning back of the NLK section to remove dry, weathered sediment, samples were
collected at intervals of 2 cm. A total of 1026 bulk samples were prepared for measurements of
physical characteristics. This study uses the more reliable optically stimulated luminescence (OSL)
dating results as basis for the age model and assessment of the evolution of loess physical
characteristics.

**3.2 Grain-size analyses**

Prior to grain size measurements, 0.5 g of dry bulk sample was pretreated by removal of organic
matter and carbonate using $H_2O_2$ and HCl, respectively (Lu and An, 1997). Samples were then
dispersed for 5 min by ultrasonification with 10 ml 10% $(NaPO_3)_6$ solution. Grain size distribution
was analysed using a Malvern 2000 laser instrument at the State Key Laboratory of Loess and
Quaternary Geology, Institute of Earth Environment, Chinese Academy of Sciences. Particle size
distribution was calculated for 100 grain size classes within a measuring range of 0.02−2000 μm.
Replicate analyses indicated an analytical error of < 2% for the mean grain size.
End-member unmixing of loess grain-size distributions is based on the hierarchical Bayesian
model for end-member modeling analysis (BEMMA) established by Yu et al. (2016). Grain-size
parameters were calculated from the analytical data with GRADISTAT (Version 4.0; Blott (2000)).
2 samples (NLK1106 at 11.06 m and NLK1840 at 17.8 m) were also selected for the extraction
of quartz grains according to published methods of Sun et al. (2000a). The isolated quartz grain
samples (Fig. S1) then placed into the Malvern 2000 laser instrument for mineral-specific grain size
measurements so that comparisons of quartz grain and bulk samples could be performed to illustrate
the weathering degree of NLK loess visually.

**3.3 Magnetic susceptibility measurements**

Magnetic susceptibility was measured with a Bartington MS2 meter at the State Key laboratory
of Loess and Quaternary Geology, Institute of Earth Environment, Chinese Academy of Sciences.
Samples were oven-dried at 40°C for 24 hours. Subsamples of 10 g from each sample were then
weighed for magnetic measurements. Low- (0.47 kHz) and high- (4.7 kHz) frequency magnetic
susceptibility ($\chi_{lf}$ and $\chi_{hf}$, respectively) were measured. The absolute frequency-dependent magnetic
susceptibility was calculated as $\chi fd = \chi_{lf} − \chi_{hf}$. Frequency-dependent magnetic susceptibility was
defined and calculated as $\chi_{fd} \% = [(\chi_{lf} − \chi_{hf})/ \chi_{lf}] \times 100\%$.

**4 Results**

**4.1 Magnetic susceptibility variations**

Both magnetic susceptibility (MS) data and stratigraphy show a close correspondence
throughout the NLK section. We observe higher MS values within primary loess and lower values
within paleosols. The exception to this trend is the modern (S0) soil in which high MS values are



presented (Fig. 2).

Fig. 2 Lithology and magnetic susceptibility characteristics ($\chi_{lf}$, $\chi_{fd}$ and $\chi_{fd}$%) of the NLK section.

The low-frequency magnetic susceptibility ($\chi_{lf}$) values of the S0 unit are higher than for the L1
unit, with an average of $98.13 \times 10^{-8} m^3 kg^{-1}$. The $\chi_{lf}$ values of the L1L1 unit vary from $56.5 - 103.9$
$\times 10^{-8} m^3 kg^{-1}$, with a decreasing trend down-profile. The $\chi_{lf}$ value abruptly decreases at c. 5 m, with
generally lower values in the L1S1 unit, averaging $62.58 \times 10^{-8} m^3 kg^{-1}$. $\chi_{lf}$ in the L1L2 unit gradually
increases down profile, with significant fluctuations in the lower part; $\chi_{lf}$ values vary from $67 -$
$102.55 \times 10^{-8} m^3 kg^{-1}$. Lower $\chi_{lf}$ values are observed in L1S1 unit with an average value of $57.99 \times$
$10^{-8} m^3 kg^{-1}$. In the L1L3 unit, the $\chi_{lf}$ values vary with greater amplitude around an average value of
$68.74 \times 10^{-8} m^3 kg^{-1}$.
Absolute frequency-dependent magnetic susceptibility ($\chi_{fd}$) values likewise vary with
stratigraphy. The S0 unit yields the highest $\chi_{fd}$ value. The L1 unit is characterized by relatively
consistent and lower $\chi_{fd}$ values. Frequency-dependent magnetic susceptibility ($\chi_{fd}$%) yields the same
trend as $\chi_{fd}$, although $\chi_{fd}$% values clearly increase in the central part of L1S2.
**4.2 Mixing model of loess grain-size distributions**
The mean grain-size distribution, and variation range of volume frequencies for each grain-
size class in the dataset, are presented in Fig. 3a. The overall grain-size frequency curve shows a
unimodal pattern, if slightly skewed towards the coarser side, with the primary mode ranging from
11.9 µm to 47.5 µm. An additional small grain size peak occurs at c. 0.4 – 2 µm. Three unmixed
end members were identified (Fig. S2), yielding fine-skewed grain-size distributions with clearly
defined modes of 47.5 µm (EM1), 33.6 µm (EM2) and 18.9 µm (EM3) (Fig. 3b).

Fig. 3 End-member modelling results of the grain-size dataset of the NLK section. (a) Mean size
distribution and range of volume frequency for each size class. (b) Modelled end-members
according to the three-end-member model (modal size: ~ 47.5 µm, ~ 33.6 µm and ~ 18.9 µm).
Size limits of clay, silt and sand fractions determined by laser particle sizer are differ from those
derived by the pipette method. The upper limits of grain-size classes used here are at 4.6/5.5 µm
for clay, 26 µm for fine silt, and 52 µm for coarse silt, as previously published by Konert and
Vandenberghe (1997). Sand is designated for particle sizes > 52 µm. Therefore, EM1 and EM2
correspond to coarse silt and EM3 to fine silt.

Fig. 4 Proportional contributions of the three end-members in the NLK section.

The proportional distribution of the end members down the section is shown in Fig. 4. In the
primary loess units (L1L1, L1L2 and L1L3), the deposits are dominated by the coarser silt EM1 and
EM2, while higher proportions of fine silt EM3 are preferentially observed within the soil horizons
(S0, L1S1 and L1S2). EM1 displays high frequency, large amplitude fluctuations down the profile,
varying between 0.09 – 0.72, and clearly dominates the primary loess units and occurs in low
proportions in the soil units (Fig. 4). EM2 shows a similar trend to EM1, but with less variability
down profile. Proportions of EM2 range between 0.11 − 0.66 with minimal fluctuations within
individual units, and proportions decrease significantly in the soil units S0 and L1S2. Proportions
of EM3 remain consistently low within the primary loess units, and increase to 0.46 and 0.8 within





the soil horizons S0 and L1S2 respectively.
**5 Discussion**
**5.1 Likely mechanisms for the enhancement of magnetic susceptibility in Ili Basin loess**
Magnetic susceptibility (MS) in loess is due to the concentration of iron-bearing magnetic
minerals within the sediment (Hambach et al., 2009;Buggle et al., 2014;Liu et al., 1999;Liu et al.,
1994;Song et al., 2010). At the broadest level, this varies between primary loess and soil horizons,
with soils generally experiencing an enrichment of magnetic minerals, and corresponding higher
MS values, than primary loess deposits (Zhou et al., 1990;Maher and Thompson, 1992;Heller and
Evans, 1995;Antoine et al., 1999;Heller and Liu, 1984;Ding et al., 2002;Forster and Heller,
1997;Buggle et al., 2009a). The formation *in situ* of < 100 nm magnetite or maghemite grains during
pedogenesis is the most widely accepted interpretation for the mechanisms of loess MS
enhancement (Nie et al., 2016). Increased precipitation is conducive to chemical weathering and
biological processes during pedogenesis. Song et al. (2010) further argued that strong pedogenesis
under warm, humid climatic conditions produces new magnetic minerals. The contrast between high
and low MS in paleosols and primary loess, respectively, has formed the basis for the stratigraphic
differentiation of loess deposits. This principle has provided the foundation for large-scale
correlations between loess deposits (Marković et al., 2015;Yang et al., 2006;Ding et al.,
2002;Marković et al., 2012;Buggle et al., 2009b;Sun et al., 2006a) and with global climatic
oscillations (Bloemendal et al., 1995;An et al., 1991;Kukla et al., 1988;Heller and Liu, 1986;Heller
and Liu, 1982), initially in the Chinese Loess Plateau deposits and increasingly worldwide.
The main MS variations in the NLK loess sequence, with the exception of the S0 unit, however,
do not occur directly in association with pedogenesis (Fig. 2). A similar case also occurs in the L1
loess layers in TLD, ZKT and AXK sections, also in the Ili valley (Fig. 1) (Jia et al., 2010;Jia et al.,
2012;Song et al., 2010). The lack of a straightforward correlation between MS, loess and paleosols
indicates that an alternative explanation for this variability must be sought. Proposed mechanisms
of variations in loess magnetic susceptibility include, in addition to pedogenesis (Zhou et al.,
1990;Maher, 1998), the dilution of relatively coarse silt with a low susceptibility (Kukla and An,
1989), sediment compression and carbonate leaching (Heller and Liu, 1984), and decomposition of
plant residues (Meng et al., 1997).
Since alternative mechanisms may have played a role in the magnetization of the Ili Basin loess
deposits, we investigated different aspects of environmental magnetic properties in order to
investigate to what degree pedogenesis or the alternative mechanisms played the more critical role
in this region.
Absolute frequency-dependent susceptibility ($\chi_{fd}$) determines the concentration of magnetic
particles within a small grain size range across the superparamagnetic (SP)/stable single domain
(SSD) boundary (Liu et al., 2012) (magnetite, < ~100 nm; maghemite, < ~20 μm). Particles with
this grain size are considered to form in situ within soils during pedogenesis (Maher and Taylor,
1988;Zhou et al., 1990), and therefore $\chi_{fd}$ can serve as a direct proxy for pedogenesis (Heller et al.,
1993;Maher and Thompson, 1995;Liu et al., 2007;Buggle et al., 2014). In the NLK section, $\chi_{fd}$
yields consistently low values throughout the sequence and indicates no clear strong pedogenesis
even in the weakly developed paleosol layers (L1S1 and L1S2). Comparison between $\chi_{lf}$ vs. $\chi_{fd}$
down profile shows no correlation between MS and SP particles (Fig, S3c). These results suggest
that SP particles played only a minor role in MS enhancement in the NLK loess.
Frequency-dependent magnetic susceptibility ($\chi_{fd}$%) is used as a proxy to determine the





contribution of SP particles to MS (Zhou et al., 1990;Liu et al., 1992). At NLK, however, we observe
consistently low $\chi_{fd}$% values in both loess and paleosol layers, with a slight increase only in the
L1S1 paleosol. This observation reinforces our interpretation that the content of SP particles is very
low, and consequently that their contribution to MS can be ignored.

247        The low proportions of SP particles in the NLK loess imply that the pseudo-single-domain

(PSD) and multi-domain (MD) magnetic grains, rather than SP grains, make the more important
contribution to magnetic enhancement of NLK loess. Since PSD and MD magnetic minerals are
difficult to produce during pedogenesis (Song et al., 2010), such minerals are more likely to be
detrital in nature, deriving from the original protolith.

Fig. 5 Comparison of different grain size fractions of NLK loess with $\chi_{lf}$ (limits of grain-size classes
after Konert and Vandenberghe (1997) ).

256        In some cases, the moist conditions typically conducive to pedogenesis, including high

precipitation and rising groundwater levels, may result in the weathering, destruction and
dissolution of the magnetic minerals maghemite and magnetite (Nawrocki et al., 1996;Cornell and
Schwertmann, 2003;Maher, 1998;Grimley and Arruda, 2007;Hu et al., 2009b;Hu et al.,
2009a;Ghafarpour et al., 2016). In such cases, a negative relationship between magnetic
susceptibility and pedogenesis can develop, in contrast to the classical situation whereby $\chi_{fd}$ is
enhanced. At NLK, however, we observe no textures caused by groundwater fluctuations, and yet
very weak pedogenesis was reflected by $\chi_{fd}$. We therefore exclude groundwater fluctuations and
high levels of precipitation as a factor in our MS characteristics at NLK.

265        Increased concentrations of coarser-grained detrital magnetic minerals, resulting from periods

of increased wind strength, may enhance overall MS values. In the wind velocity/vigor model (also
known as the Alaskan or Siberian model), wind strength affects magnetic susceptibility values of
loess through the physical sorting of magnetic grains (Beget and Hawkins, 1989). The influence of
this process on MS values in loess can be assessed by investigating the correlation between MS and
coarser (silt or sand) and finer clay percentages (Fig. S3). At NLK, low MS values in the S0 soil
between 0 – 0.5 m correlate positively with clay percentage variations (Fig. S3a), while higher MS
values at depths greater than > 0.5 m correlate closely with increased sand concentrations (Fig. S3b).
We therefore propose that MS enhancement at NLK is primarily driven by increased concentrations
of sand-sized detrital magnetic minerals, which increase during periods of stronger winds. The
dilution effect of coarse particles with low susceptibility was excluded.

276        In the case of NLK, the reduced color contrast (Fig. 1) between loess and paleosol layers

implies moderate climate fluctuations between loess deposition and pedogenesis due to generally
more arid conditions than typically experienced in loess regions. This prevents the efficient
production of SP grains (Fig. 2). Wind strength can therefore be regarded as a main factor for MS
variations since last glacial. And in turn, MS may be able to indicate stronger wind during dust
storms.

**282    5.2 Genetic interpretations of end members in loess grain size**

283        In order to understand the atmospheric dynamic pattern during loess deposition further, we

conducted unmixing of grain-size distributions.

285        Recent years have seen increasing statistical analysis of loess grain-size to identify

subpopulations within bulk samples (Prins, 2007;Prins et al., 2007;Qin et al., 2005;Sun et al.,





2002;Vriend et al., 2011;Vandenberghe, 2013;Sun, 2004). From these statistical datasets, the
different end members can be interpreted to infer distinct atmospheric transport mechanisms, modes
and travel distances (Ujvari et al., 2016). In some cases, the end-member approach has been used to
identify variation in geological context, or source area (Prins et al., 2007). We investigated the utility
of this approach to the Ili Basin loess at NLK by unmixing grain-size distributions with BEMMA
(Yu et al., 2016). As shown in Fig. S2, we generated a mixing model consisting of three end
members.
Fine sand ('*sediment type 1.a*' in Vandenberghe (2013)) is a typical component of loess deposits
near to or overlying river terraces. Although the NLK section lies on the second terrace of the Kashi
River and therefore closer to a potential source of coarser grained material, the fine-sand end
member is completely absent. Modal grain sizes in this range (c. 75 um) are common in loess along
the Huang Shui and Yellow Rivers in China (Vriend and Prins, 2005;Vandenberghe et al., 2006;Prins
et al., 2009), the Danube and Tisza rivers in Serbia (Bokhorst et al., 2011), and the Mississippi valley
in the USA (Jacobs et al., 2011). This fraction is generally interpreted to originate from proximal
sources, and the grain size of the available source material plays a more important role in
determining the grain-size characteristics of this fraction than wind energy (Vandenberghe, 2013).
The lack of fine sand at NLK, despite its proximity to the Kashi River, may be attributed to 1) its
location in the upper reaches of the river (Fig. 1b), in a region which lacks available stocks of fine
sand, 2) the V-shaped nature of the channel which is not conducive to aeolian transport of bank
deposits, and 3) the relatively high altitude of NLK within the basin which inhibits transport and
deposition of coarser sediment grains (Vandenberghe, 2013).
The three members (Fig. 3b) identified at NLK correspond to coarse silt (EM1 and EM2) and
fine silt (EM3). Each likely represent different kinds of depositional processes which operated
throughout the accumulation of the deposit at NLK. Here we focus on the implications of these three
end members for understanding past environmental conditions responsible for loess-paleosol
sequences formation.
EM1 has a modal grain size of 47.5 µm (Fig. 3b), which approximately corresponds to the
'*subgroup 1.b.1*' of Vandenberghe (2013). The mode is similar to end members identified in loess
from the Chinese Loess Plateau (CLP) and the north-eastern Tibetan Plateau (NE-TP) (EM-2: 44
µm) (Vriend et al., 2011). The size of this component is unlikely to be due to longer distance
transport. Therefore it is inferred that EM1 is derived from shorter distance transport of suspended
load (Vriend et al., 2011;Vandenberghe et al., 2006). Coarser particles with grain-size >20 um rarely
reach suspension above the near surface level (0 − 200 m above the ground). When entrained by
wind, they do not remain in suspension for long enough to travel long distances (Tsoar and Pye,
1987;Pye, 1987). Since the average grain-size of EM1 is 26.74 µm (calculated after Folk and Ward
(1957)), we infer that this fraction was transported mainly in short-term suspension episodes at
lower elevations by surface winds, and deposited short distances downwind of the source. These
short-term suspension episodes may correspond to spring-summer dust storms, as demonstrated by
present-day dust measurements on the CLP which detected a similar modal grain-size during these
events (Sun et al 2003).
EM2 represents a mode at 33.6 µm (Fig. 3b). It lies towards the finer end of the range of
'*subgroup 1.b.2*' (Vandenberghe, 2013). Comparable loess of the same grain size has been identified
in loess from the northern Qilian Shan/Hexi Corridor (EM-2: 33 µm), which was also interpreted as
depositing from short-term suspension (Nottebaum et al., 2015). Loess of this grain size has been



attributed to dust fallout (Pye, 1995;Muhs and Bettis, 2003) and from low-altitude suspension clouds
(Sun et al., 2003), as measured from modern depositional events. This fraction requires less wind
energy than EM1, is transported further, is more widely distributed, and therefore comprises a higher
proportion of the distally deposited population in loess generally (Vandenberghe, 2013). We propose
that EM2 was transported mainly in short-term, near-surface suspension during dust storms, and
that wind strength controlled the relative proportions of EM1 and EM2 through time (see the mirror
image relationships over millennial scales in Fig. 4), which may implied that both EM1 and EM2
have a same origin.
The grain-size distribution of EM3 has a modal peak at 18.9 µm (Fig. 3b). This population
belongs to '*subgroup 1.c.1*' in Vandenberghe (2013). This population is also widespread in loess
from the CLP and northeastern Tibetan Plateau (Prins et al., 2007;Prins, 2007), and the Danube
Basin loess of Europe (Bokhorst et al., 2011;Varga, 2011), particularly in loess of interglacial age
(Vriend, 2007). There is as yet no consensus as to the transport processes responsible for this grain
size population. On the one hand, researchers have suggested that grains of this size can be lifted by
strong vertical air movement and subsequently incorporated into the high-level westerly air streams
(Pye, 1995;Pye and Zhou, 1989). This process would link EM3 with long-term suspension transport
driven by high-level Westerlies (Prins et al., 2007;Vriend et al., 2011;Nottebaum et al.,
2014;Vandenberghe, 2013). Conversely, Zhang et al. (1999) argued that EM3 derives from "non-
dust storm processes" associated with north-westerly surface winds. We argue for the latter on the
basis that the EM3 modal grain size from the CLP and northeast Tibetan Plateau is coarser (Vriend
et al., 2011) than EM3 at NLK in the Ili Valley, which is located further west. If EM3 was transported
by high-level westerlies, then one would expect either no significant change (Rea et al., 1985;Rea
and Hovan, 1995), or a decrease in grain size from west to east concomitant with wind direction.
Furthermore, with mathematical fitting, Sun et al. (2004) related a fine component (2 – 8 µm) to
high-altitude westerlies. This fine component is comparable to '*subgroup 1.c.2*' of Vandenberghe
(2013), which is not consistent with the modal size of EM3. Observations of modern aeolian
processes at the southern margins of the Tarim Basin indicate that fine grain sizes similar to EM3
(8 − 15um) are deposited by settling during low velocity wind conditions (Lin et al., 2016). We
therefore infer the EM3 modal peak to derive from low altitude non-dust storm processes.
Other possibilities for the deposition of the fine particles include the incorporation into silt- or
sand-sized aggregates which can be transported by a range of wind velocities including dust storms
(Qiang et al., 2010;Pye, 1995;Derbyshire et al., 1998;Mason et al., 2003). For example, Ujvari et al.
(2016) indicated that the ~ 1 – 20 µm fractions are affected by aggregation by comparison between
minimally and fully dispersed grain size distributions of loess samples from southern Hungary.
Under higher wind velocity conditions, the aggregate model should co-vary with the coarser EM1
particles which were transported by surface winds during dust storms. However, since this model is
unlikely to hold for EM3 particles (Fig. 4), the aggregate model is not thought to be responsible for
the presence of grain sizes corresponding to EM3.
In addition, post-depositional processes may also influence grain size distribution. In large part
this occurs by chemical weathering which produces very fine silt and clay minerals (Xiao et al.,
1995;Wang et al., 2006;Hao et al., 2008). In particular, quartz grains are more weathering resistant
and remain largely unaltered during the post-depositional processes. Consequently, quartz mineral
grain size may be used as a more reliable proxy indicator of winter monsoon strength than other
components (Sun et al., 2006b;Sun et al., 2000b;Xiao et al., 1995).





Figure. 6a shows the grain size distribution curves of quartz grains isolated from primary loess
(yellow) and paleosol (red) samples. The quartz modal grain size is finer in the paleosol than in the
primary loess unit. From this we can deduce that wind strength was weaker during pedogenesis, and
stronger during periods of primary loess deposition. The grain size distributions of bulk samples
display similar characteristics with those of quartz samples mentioned above (Fig. 6b), whereby soil
unit modal peaks (red and orange) are finer than those for primary loess (blue and green). Therefore,
we argue that wind strength, rather than the post-depositional pedogenesis, has the greatest influence
on grain size distribution at NLK, and that EM3 was also not produced by chemical weathering.
Fig. 6 Comparison of grain size distribution between purified quartz subsamples of paleosol and
primary loess (a), and between bulk samples of paleosols and primary loess (b). Comparison of
the grain size distribution between EM3 and samples from weak paleosol units (c).
The relative proportions of the end members down profile can yield further information about
temporal variability in wind dynamics. The fairly consistent proportions of EM3 within the loess
units indicate it to represent continuous background dust through time (Vandenberghe, 2013).
Proportions of EM1 and EM2 decrease noticeably within paleosol units relative to EM3 (Fig. 4).
This indicates that variations in proportions of EM3 are mainly driven by variability of EM1 and
EM2 (Vriend et al., 2011), but also that a background sedimentation of EM3 was dominant during
weak pedogenesis (Fig. 6c). This characteristic is comparable with observations from the CLP
(Zhang et al., 1999).
In addition, small peaks at c. 0.8 µm are also observed in the grain-size distribution curves of
all three end members. The generation of finest grain peaks may be due to post-depositional
pedogenesis (Sun, 2006), especially for the particles with grain-size smaller than 2 µm (Bronger
and Heinkele, 1990;Sun, 2006). Nevertheless, post-depositional weathering is unlikely to have had
a significant influence on the populations of EM1, EM2 or EM3, since the dominant modal peaks
are much coarser. Other potential sources include transportation as aggregates or by the finest grains
adhering to coarser particles during transport. Regardless of cause, these particles are unlikely to
yield meaningful information about variability in westerly wind system strength since they do not
yield a clear independent end member peak.

**5.3 Aeolian dust dynamics in eastern Central Asia: links to atmospheric systems**

Variations in grain size through time at NLK were largely driven by changes in wind strength,
without substantial influence of post-depositional pedogenesis. At NLK, grain size therefore is an
indicator for response to the atmospheric system.
The three end members are interpreted to represent different depositional processes which
operated throughout the accumulation of the deposit. The finer EM3 is interpreted to represent
constant background dust, which continued to accumulate throughout periods of relative stability
and pedogenesis. The coarser populations, EM1 and EM2, were transported by low-level winds
during major dust storms. EM1 is most likely the most sensitive recorder of wind intensity, since
EM2 is less sensitive to wind speeds than EM1 by observation of variations in EM2 proportions
throughout L1S1 and L1L2 (Fig. 4).
Fig. 7 Comparison between EM1 grain size variability with the timing of glacial advances in the
Tian Shan (Koppes et al. 2008; Owen and Dortch, 2014); stable oxygen isotope variations from the



Greenland ice cores (Rasmussen et al., 2014) and insolation values at 45°N (Berger and Loutre,
420 1991).
From the OSL data (Song et al., 2015), we used linear regression (Stevens et al., 2016) to
construct age–depth relationships over intervals of visually similar sedimentation rate (Fig. S4 and
Table S1). Based on the independent chronology sequences, we assess the degree of correlation
between wind strength variability in the Ili Valley (NLK), as represented by the proportions of EM1,
the stable oxygen isotope record from the Greenland ice cores representing North Atlantic
paleoclimate (Rasmussen et al., 2014), insolation values at 45°N (Berger and Loutre, 1991) and
glacial advances in the Tian Shan (Owen and Dortch, 2014;Koppes et al., 2008) (Fig. 7).
In Fig. 7, EM1 occurs in larger proportions during mid-MIS3, with a higher rate of sedimentary
accumulation. Glaciers expanded during early- and late-MIS3 (Owen and Dortch, 2014). Generally
dust is assumed to be generated, and deposited, during dry-windy glacial conditions, while
interglacial conditions were comparatively wetter and more conductive to pedogenesis (Stevens et
al., 2013;Sun et al., 2010;Ding et al., 2002;Dodonov and Baiguzina, 1995). By contrast, a seesaw
relationship between rapid loess deposition and glacial expansion was observed during MIS3 from
our results (Fig. 7), a model that has also been noticed by Youn et al. (2014). The mass accumulation
rate (MAR) of loess is good proxy for aridity (Pye, 1995), while moisture availability is the
dominant factor controlling glacier growth in Central Asia, especially for glaciers in the Tian Shan
(Zech, 2012;Koppes et al., 2008). We infer, therefore, that moisture had an important impact on
accumulation of dust in the study area over MIS3 in particular.
Central Asia is variably influenced by the Asian monsoon from the south (Dettman et al.,
2001;Cheng et al., 2012), the mid-latitude westerlies (Vandenberghe et al., 2006), the Siberian high-
pressure systems from the northeast (Youn et al., 2014), and the polar front from the north
(Machalett et al., 2008). However, by virtue of its geographical position, most of these climate
influences can be excluded for the Ili Valley since it is sheltered to the northeast, east and south..
The Asian high mountains largely inhibit the intrusion of Asian (Indian and East Asian) monsoons
to the region, and the influence of the Siberian High (An, 2000) has been shown to decrease
westward from the CLP (Vandenberghe et al., 2006).
Modern satellite data indicates that dust storm development in Ili river valley is closely linked
with southward-moving high-latitude air masses (Ye et al., 2003). Karger et al. (2016) provided a
detailed picture of the westerlies for the Ili Basin, in which a rain belt gradually migrated towards
the south and north in autumn and summer, respectively. According to this scenario, enhanced
evaporation coupled with strengthened westerly winds would bring more humid and warm air
masses to Arid Central Asia (ACA) during the Holocene (Zhang et al., 2016). Therefore, based on
our grain-size observations, we argue that the Arctic polar front, intruding southward in the winter
and retracting northward in summer (Machalett et al., 2008), most likely increased the frequency
and strength of cyclonic storms, leading to dust transport and the accumulation of loess deposits
during cold phases when it predominated in the Ili Basin and along the Kyrgyz Tian Shan piedmont.
While the mid-latitude westerlies increasingly influenced the climate in this region as the climate
became warmer when the polar front shifted northward, and controlled the patterns of moisture
variations (Huang et al., 2015;Li et al., 2011).
Comparison of EM1 proportions with variability in June insolation at 45°N shows a distinct
correlative relationship on the orbital timescale (Fig. 7), which indicates local insolation-based





control on wind dynamics. When the insolation values increases, the rising of temperature, as a
result, enhances the frequency or strength of cyclonic storms, resulting in higher sedimentary rates
or higher coarse-grain proportions (Fig. 7). However, EM1 proportions exhibit more substantial
fluctuations than may be attributed to insolation values during the mid- and late-MIS3. We ascribe
that to the humidity variations in the study area. In the early-MIS 3, increased moistures due to
migration of westerlies towards the north were conducive to vegetation growth in source areas,
which reduced sediment entrainment and resulted in less contribution of coarse grains to loess site,
though glacial grinding of rocks in the high mountains could produce amount of fine-grained
materials (Smalley, 1995;Li et al., 2016a;Fitzsimmons et al., 2016). Whereas arid environment in
the mid-MIS 3, observed by a lack of glacial advance in Tian Shan (Fig. 7) and also reflected by the
increased MAR (Fig. 7) (Pye, 1995), likely made these sediments with coarser grain size produced
in the early-MIS 3 available as the source materials for NLK loess, as the case in the north-eastern
Tibetan Plateau (Vriend et al., 2011).

Over millennial scales, our grain-size proxy data do not correlate strongly with abrupt events,
such as H1, H2, H3 and H5, identified from the North Atlantic records (Fig. 7). Some of the peaks
in EM1 curve correspond to valleys in GISP $\delta^{18}O$ curve (black arrows in Fig. 7), yet many do not.

Grain size studies of the Darai Kalon loess section in Tajikistan, 1200 km to the southwest of
NLK, inferred a strong influence from the westerlies resulting in transport of the North Atlantic
signal to the East Asia (Vandenberghe et al., 2006;Porter and An, 1995;Sun et al., 2012). Darai
Kalon is, however, located in a region where the mid-latitude westerlies clearly have a much
stronger influence. Our results from the Ili Basin contradict those of Vandenberghe et al. (2006),
which suggest that the mid-latitude westerlies probably did not predominate north of the Kyrgyz
Tian Shan. In this case, the high mountains in Central Asia most likely obstructed the migration of
the Asiatic polar front further south towards Tajikistan where those data were derived, thereby
resulting in a stronger westerlies signal at Darai Kalon than at NLK.

Our results also contradict those of Yang and Ding (2014), who proposed that millennial-scale
North Atlantic climate signals might have been transmitted to the Siberian High via the Barents and
Kara Sea ice sheets, and then propagated eastwards to the Chinese Loess Plateau via the winter
monsoon system. In our case, the influence from northern climate subsystems such as the Siberian
High or polar front appear not to have transmitted millennial-scale North Atlantic climatic events,
maybe supporting the significance of the westerlies in transmitting North Atlantic climate signals
to East Asia.

**Conclusion**

In this study, a paleoenvironmental record for the last glacial from the Nilka (NLK) loess
section in Ili Basin was provided. The magnetic properties of the loess indicate that no strong
pedogenesis occurred in this section, even in the paleosol units. Variations in magnetic susceptibility
(MS) value closely correlate with the proportions of sand fraction, and wind strength is mainly
responsible for those variations since the last glacial.

With the unmixing of grain size distributions, three end members were distinguished: EM1
(mode size at 47.5 µm), EM2 (33.6 µm) and EM3 (18.9 µm). They are indicative of different kinds
of depositional processes which operated throughout the accumulation of the loess deposit at NLK.
EM1 and EM2 represented the grain-size fractions transported from proximal sources in short-term,
near-surface suspension during dust outbreaks. They may have the same origin. While wind strength
controls relative proportions, EM1 is most likely the most sensitive recorder of wind strength. EM3



represents continuous background dust under the non-dust storm processes.
The Arctic polar front predominates in the Ili Basin and the Kyrgyz Tian Shan piedmont during
cold phases, which leads to the dust transport and increased accumulation of loess deposits, while
the shift of mid-latitude westerlies towards the south and north controlled the patterns of
precipitation/moisture variations in this region. On the orbital scale, the local insolation-based
control has an important impact on wind dynamics directly related to accumulation of loess, and
moisture can may also influence grain size of loess in the study area over MIS3 in particular.
Although, the polar front dominated wind dynamics for loess deposition in the Ili Basin and the
Kyrgyz Tian Shan, the Central Asian high mountains obstructed its migration further south. Our
results may also support the significance of the mid-latitude westerlies in transmitting North Atlantic
climate signals to East Asia.
**Acknowledgements**
The project is supported by the National Basic Research Program of China (Nos:
2016YFA0601902, 2013CB955904), Natural Science Foundation of China (Nos: 41572162,
41290253), and International partnership Program of Chinese Academy of Science [No:
132B61KYS20160002]. The authors thank Yun Li and Junchao Dong from Institute of Earth
Environment, Chinese Academy of Sciences, for their assistances in sampling and experiment, and
Jia Li from Xiamen University for her assistance in mathematical treatment.

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





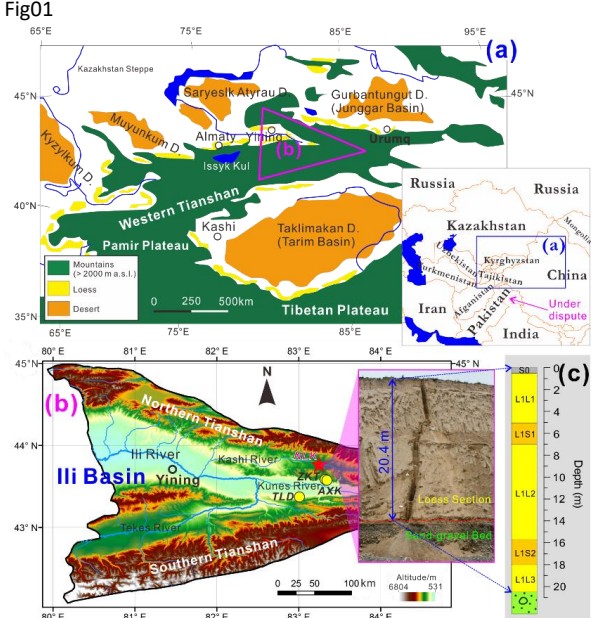



Fig02

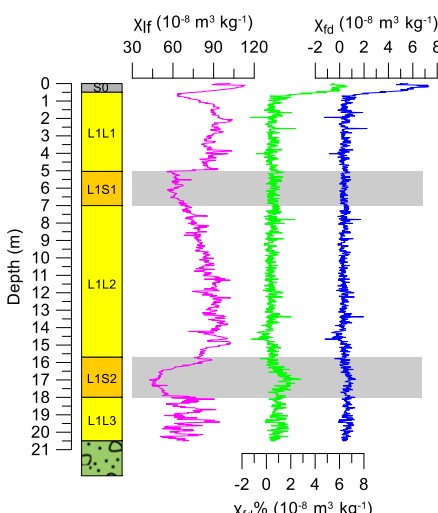



Fig03

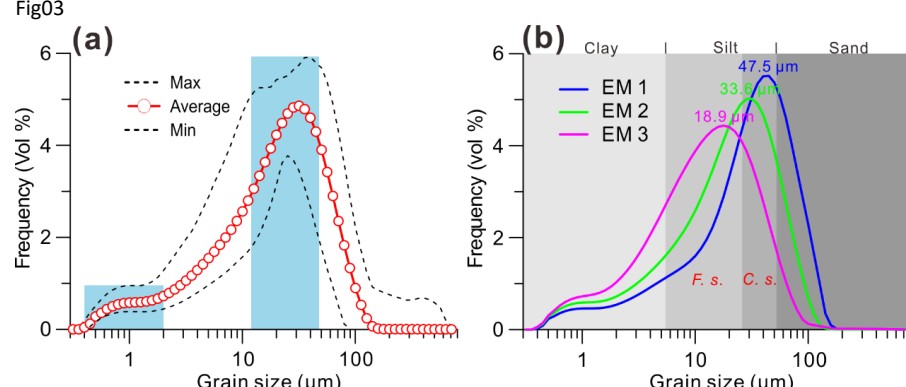




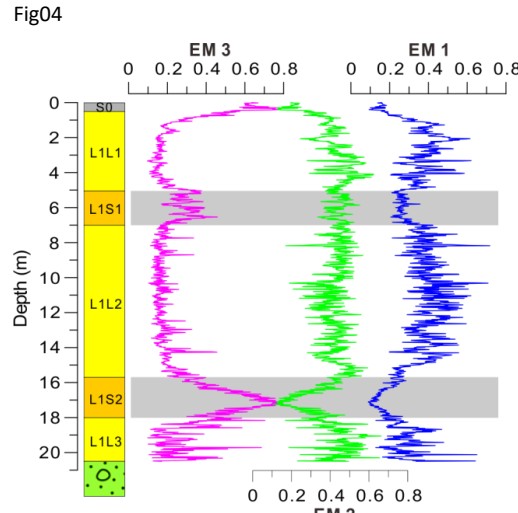





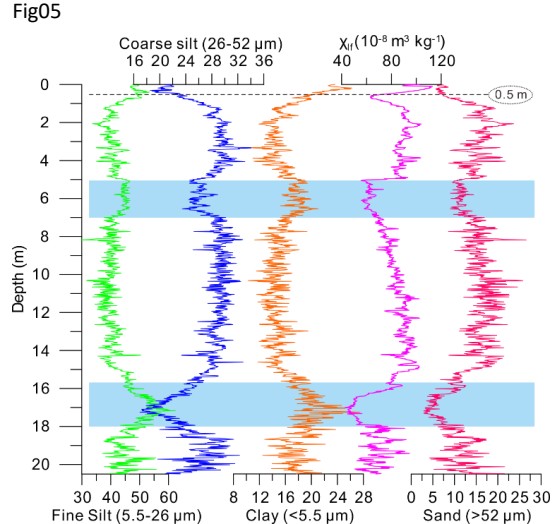

Fig05





Fig06

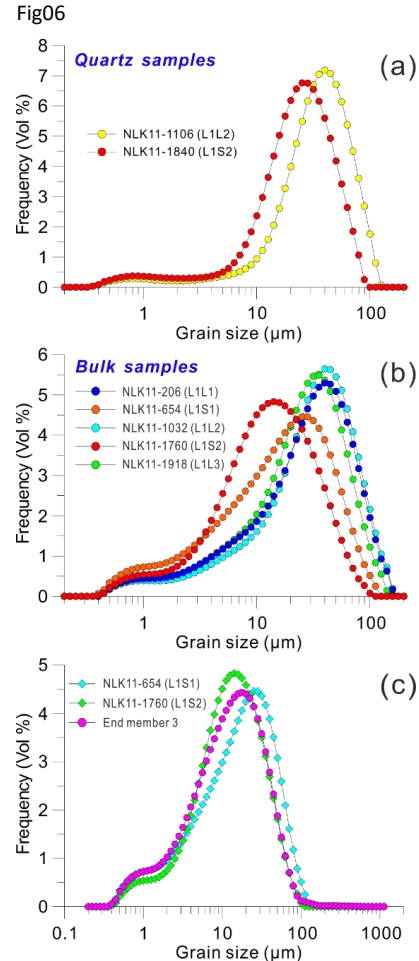




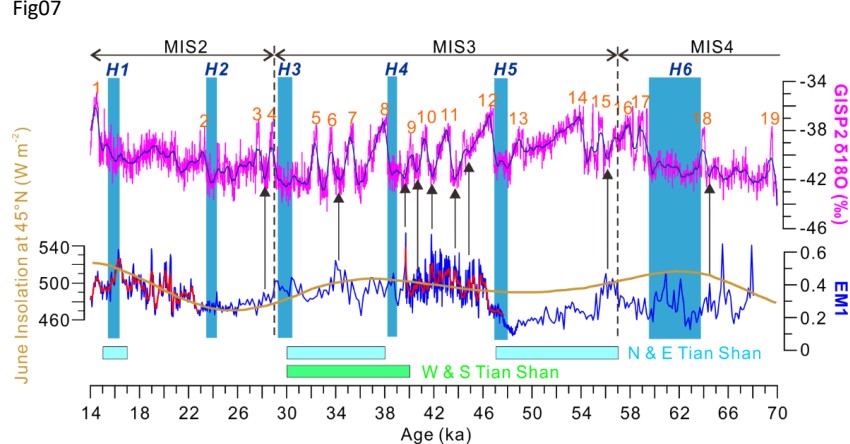