# Peer review of "1. Separation of quartz minerals from loess samples"

_Climate of the Past, 2017_

## Short Comment (SC1) · 14 Apr 2017

General Comments: Many studies deal with the correspondence between loess deposits and climate circulation but the causal relations are indeed poorly understood. Previously, the attention has been drawn to competing circulation systems of westerlies and monsoons. Thus, relations are indeed best studied in a region where different systems could have been present at times. Consequently, the studied region is favorably situated for this timely research. A most interesting result is the importance of precipitation on the loess depositional signal. Until now most attention has been paid

to temperature variability that of course also impacts the wind circulation. It seems by this study that resulting oscillations in loess deposition show a complex pattern. And this may be the reason for the fact that the oscillating loess signals in C. Asia and the NE Tibetan Plateau are not that simply correlated with D-O-events or H-events. The paper is well written, designed and archived.

Minor comments -l 343-359: Two different explanations are claimed for the origin of the fine-grained endmember 3 (c. 18.9 $\mu$m). The main difference seems to be, if I understand well, that one hypothesis invokes high-suspension transport while in the other one surface winds are involved. However, both hypotheses interpret that this component is the result of background loess supply (as confirmed in lines 389-395) as previously demonstrated by Prins et al (2007), Vriend et al (2011) and Zhang et al ( (1999). It is not realistic to separate the grain-size fractions of 2-8 $\mu$m (transported by westerlies) and 8-15 $\mu$m (=EM3, transported at low altitude) as the authors seem to do. Both components react jointly constituting background loess supplied by westerlies as described by e.g. Prins eta l. (2007). -l 441-447: If the Ili valley is sheltered from northeastern wind, as the authors claim, what is then the source area for the EM1 and EM2 fractions? There is no apparent difference between these coarse-grained fractions on the CLP, N Tibet Plateau and in the Ili valley where a distinct supply is clear from the northeast under the influence of the Siberian High. -l 454-456: Explain better how the 'cyclonic storms' originated by protrusion of the Arctic polar front, rather than by other circulation patterns. -l 476: The interesting absence of correlation between the observed grain-size signals and N Atlantic abrupt events is not only found in the Ili valley but also previously in Tadjikistan and the NE Tibet Plateau (Vandenberghe et al. 2006). -l 483-484: This sentence is not clear: is 'which' referring to the conclusions by the authors or by Vandenberghe et al.? It is not clear therefore what really is contradicting.

Technical comments: L 123: 'more reliable' than what? L139: insert 'were' between 'S1)' and 'then'; Figure 1 is too small. L 182: remove 'are'; L 317: 'shorter' than what?

L 513: remove 'can' or 'may'. I suggest to shorten the title a bit.

Jef Vandenberghe

---

## Referee Comment (RC2) · J. Vandenberghe (Referee) · 5 May 2017

Dear authors, I agree with most of your replies and thank you for the modifications. I just want to react with 2 comments: 1. To the origin of the very fine silt-clay component: Chemical weathering is indeed a good candidate as measured by Konert and Vandenberghe 1997, and well illustrated by the experiments of Sun YB et al 2006. Transport as aggregates of fines by monsoonal dust storms (Qiang et al 2010) is contradicted by their very widespread and general occurrence (Vandenberhe 2013. Adherence of fines to larger grains has been contradicted by several authors. 2. Provenance of EM 1-2:
I agree with your explanation. I understand now that you also agree with a northern wind, however not crossing the high mountains to the north but carrying dust only at low elevation over short distance. In my opinion, the carrying agent may still be the northern monsoonal wind, although restricted to the Ili basin. Jef Vandenberghe

———————————————————

---

## Author Comment (AC1) · 5 May 2017

For General Comments

CS: Many studies deal with the correspondence between loess deposits and climate circulation but the causal relations are indeed poorly understood. Previously, the attention has been drawn to competing circulation systems of westerlies and monsoons. Thus, relations are indeed best studied in a region where different systems could have been present at times. Consequently, the studied region is favorably situated for this timely research. A most interesting result is the importance of precipitation on the loess

depositional signal. Until now most attention has been paid to temperature variability that of course also impacts the wind circulation. It seems by this study that resulting oscillations in loess deposition show a complex pattern. And this may be the reason for the fact that the oscillating loess signals in C. Asia and the NE Tibetan Plateau are not that simply correlated with D-O-events or H-events. The paper is well written, designed and archived.

AR: Thanks for your positive comments. This paper made aims to investigate the causal relations between loess deposits and climate circulation. Recently, many researchers have paid much attention to paleoclimate reconstruction in transitional regions where different atmospheric systems predominate at different times, e.g. the Central Balkans (Ramisch et al., 2016), the Qinghai Lake region (An et al., 2012), southwest Asia (Hamzeh et al., 2015). Our study area is likewise situated in the zone influenced by different climatic systems. It therefore becomes important prerequisite to clarify the paleoenvironmental significance of various proxies, in particular which proxies reflect temperature, which represent precipitation, and which indicate wind strength. Our manuscript explores the potential impacts of precipitation on the loess depositional signal, and identifies areas where more work needs to be done in the future. We agree that some scientists have focused on the influence of temperature variability on wind circulation, and in particular the role of local insolation minima in driving an early onset of the LGM in the Southern Hemisphere (Vandergoes et al., 2005). The temperature variability caused by local insolation may reflect larger scale climate change, although we suspect that oscillating loess signals in Central Asia may not so simply reflect D/O events or H events. The aeolian loess sediments in Central Asia are more likely to respond to a complex mixture of global signals with local insolation, glacial activity and local weathering. This overlap will weaken the global signals as preserved within the loess.

For Minor Comments

CS: L 343-359: Two different explanations are claimed for the origin of the fine-grained

endmember 3 (c. 18.9 $\mu$m). The main difference seems to be, if I understand well, that one hypothesis invokes high-suspension transport while in the other one surface winds are involved. However, both hypotheses interpret that this component is the result of background loess supply (as confirmed in lines 389-395) as previously demonstrated by Prins et al (2007), Vriend et al (2011) and Zhang et al (1999). It is not realistic to separate the grain-size fractions of 2-8 $\mu$m (transported by westerlies) and 8-15 $\mu$m (=EM3, transported at low altitude) as the authors seem to do. Both components react jointly constituting background loess supplied by westerlies as described by e.g. Prins et al. (2007).

AR: We agree with the reviewer on this point. The origins of the EM3 size fraction is indeed complex. For example, based on modern dust monitoring from the high-altitude subtropical Puna-Altiplano Plateau in South America, Gaiero et al. (2013) found that "Finer mode dust is deposited during event periods, which point to a dominant long-range transport, contrasting with a dominance of coarser mode observed for non-dust sampling periods, pointing to dominant local sources." Prins and Vriend (2007) and Prins et al. (2007) suggested that the clayed loess component represented the fine dust component supplied over the entire Loess Plateau by long-term suspension processes, and the high-level subtropical jet stream (westerly winds) might, at least partly, be responsible for the input of this fine-grained loess component. End-member unmixing results of Xiaoerbulake (XEBLK) loess (Li et al., 2016) grain-size distributions show the similar EM3 component to NLK loess (Fig. 1), which suggests that the fine-grained EM3 (c. 18.9 $\mu$m) is also the result of background loess supply in the Ili Basin regardless of its origin (Vriend et al., 2011;Zhang et al., 1999;Prins et al., 2007). It is difficult to determine the origins of the fine silt/clay. The appearance of the fine component in dust deposition may be caused by aggregation, due to fine particles adhering to the coarse particles, as well as chemical weathering. Perhaps the method of Machalett et al. (2008) is the better alternative. They neither removed organic matter and carbonates from the stratigraphic samples and nor applied an intensive ultrasonic treatment to disaggregate particles.

CS: L 441-447: If the Ili valley is sheltered from northeastern wind, as the authors claim, what is then the source area for the EM1 and EM2 fractions? There is no apparent difference between these coarse-grained fractions on the CLP, N Tibet Plateau and in the Ili valley where a distinct supply is clear from the northeast under the influence of the Siberian High.

AR: This is a constructive question. The northern Tien Shan Range reaches altitudes of > 4000 m a.s.l. For the particles with grain size of > 20 $\mu$m, it is unlikely that grains of this coarser silt fraction were transported by north-easterly winds above the 4000 m altitude over the northern Tien Shan and into the Ili Basin. We therefore interpret the coarser grained loess particles in the Ili Basin to have been predominantly transported by near-surface winds. The topographic context (see Fig. 1 in the manuscript) most likely ensured the westerly winds coming to be the transporting agent. In our speculation as to the provenance of the NLK loess, we initially compared the REE parameters of NLK loess with those of desert sands and modern soils from the Ili Basin and further west into Kazakhstan (Fig. 2). Our results indicated that the deserts and topsoils in Kazakhstan are unlikely to be the main potential source areas. In contrast, topsoils from the Ili Basin probably provide the most important source materials in the NLK loess. The Quaternary sediments of the Ili Basin mainly consist of alluvial fans and floodplains, and the top soils developed on those. We therefore speculate a proximal source for the NLK loess. Furthermore, recent work from our group indicates that size-differentiated rare earth elements (REE) may help to distinguish potential proximal or distal sources (Chen et al., 2017). In future, we expect to find more substantial evidence for tracing loess provenance in the region.

CS: L 454-456: Explain better how the 'cyclonic storms' originated by protrusion of the Arctic polar front, rather than by other circulation patterns.

AR: Thank you for this suggestion. After excluding the influences of the Asian monsoon and the Siberian high-pressure system based on modern observations, we consider that both the polar front and mid-latitude westerlies had important impacts on

our study area. Cyclonic circulation associated with the polar front is one of the major driving forces for aeolian dust transport in central Asia today (Lydolph, 1977;Liu et al., 2004). The importance of the polar front is also embodied in other studies (Kirby et al., 2002;Diefendorf et al., 2006;Schöne et al., 2004). According to Harman (1991) and Shriner and Street (1998), the polar front is a discontinuous border zone that generally separates the moister tropical air to the south from the drier polar air to the north. These two air masses from the south and north can generate temperature and pressure differences, which will be settled by the development of the near-surface midlatitude cyclones and the development of troughs in the polar front jet stream. We therefore speculate near-surface northerly transport of the NLK loess deposits with a mean grain size of $> 20~\mu$m, whereas westerlies occur in the upper atmosphere throughout the year and are responsible for finer grained transport (Fig. 3). Therefore, we link the polar front with loess deposition at NLK. When the polar front moved southwards in the cold periods, the frequency and strength of cyclonic storms were most likely increased, leading to dust transport and the accumulation of loess deposits (Fig. 8 in Machalett et al. (2008)).

CS: L 476: The interesting absence of correlation between the observed grain-size signals and N Atlantic abrupt events is not only found in the Ili valley but also previously in Tadjikistan and the NE Tibet Plateau (Vandenberghe et al. 2006)

AR: That point has been attracting the attention of our group recently. In our view, the lack of good correlation between observed grain-size and millennial-scale Atlantic events suggests that the loess records in Central Asia represent a response not only to global signals but also local signals such insolation, glacial activity and local weathering. This overlap will weaken the global signals. In contrast, on the orbital timescale, the mid-westerlies may impact on the accumulation of loess deposits by controlling precipitation patterns owing to northward or southward migration of the system.

CS: L 483-484: This sentence is not clear: is 'which' referring to the conclusions by the authors or by Vandenberghe et al.? It is not clear therefore what really is contradicting

AR: I am so sorry for the poor expression. The 'which' referred to the conclusions by the authors. We have revised the sentence, like this "Darai Kalon is located in a region where the mid-latitude westerlies clearly have a much stronger influence, especially during full glacial conditions (Vandenberghe et al., 2006). In contrast, our results from the Ili Basin suggest that the mid-latitude westerlies did not always predominate north of the Kyrgyz Tian Shan due to northward or southward movement of the climate sub-system. In this case, the high mountains in Central Asia most likely obstructed the migration of the Asiatic polar front further south towards Tajikistan where those data were derived (Machalett et al., 2008), thereby resulting in a stronger westerlies signal at Darai Kalon than at NLK." The movement northward or southward of mid-latitude westerlies makes the Ili Basin more sensitive to paleoclimate change in Central Asia, which establishes the strategic position of the Ili Basin in paleoclimatic reconstruction.

For Technical comments

CS: L 123: 'more reliable' than what?

AR: Thank you for your careful reading. We have rewritten this sentence. Actually, we mean that the optically stimulated luminescence (OSL) dating is more reliable for constructing a loess chronology than AMS 14C ages for older than MIS2 aeolian sediments according to Song et al. (2015).

CS: L 317: 'shorter' than what?

AR: EM1 is likely derived from shorter distance transport of suspended load owing to its larger modal grain size. Thus, its transport distance is shorter than the finer grains, like the EM2 and EM3 fractions in this manuscript.

CS: L 139: insert 'were' between 'S1)' and 'then'; Figure 1 is too small. L 182: remove 'are'; L 513: remove 'can' or 'may'.

AR: Yes, these are grammar errors. We have corrected these mistakes accordingly. We also adjusted the layout of Fig. 1 and increased front size. Thank you.

CS: I suggest to shorten the title a bit

AR: Yes, we have rewritten the title, "Environmental dynamics since the last glacial period in arid Central Asia inferred from loess deposits in the Ili Basin". We look forward to your further comments about that. Thank you.

References

An, Z., Colman, S. M., Zhou, W., Li, X., Brown, E. T., Jull, A. J. T., Cai, Y., Huang, Y., Lu, X., Chang, H., Song, Y., Sun, Y., Xu, H., Liu, W., Jin, Z., Liu, X., Cheng, P., Liu, Y., Ai, L., Li, X., Liu, X., Yan, L., Shi, Z., Wang, X., Wu, F., Qiang, X., Dong, J., Lu, F., and Xu, X.: Interplay between the Westerlies and Asian monsoon recorded in Lake Qinghai sediments since 32 ka, Scientific reports, 2, 619, 2012. Chen, X., Song, Y., Li, J., Fang, H., Li, Z., Liu, X., Li, Y., and Orozbaevd, R.: Size-differentiated REE characteristics and environmental significance of aeolian sediments in the Ili Basin of Xinjiang, NW China, Journal of Asian Earth Sciences, 143, 30-38, 2017. Diefendorf, A. F., Patterson, W. P., Mullins, H. T., Tibert, N., and Martini, A.: Evidence for high-frequency late Glacial to mid-Holocene (16,800 to 5500 cal yr BP) climate variability from oxygen isotope values of Lough Inchiquin, Ireland, Quaternary Research, 65, 78-86, 2006. Gaiero, D. M., Simonella, L., Gassó, S., Gili, S., Stein, A. F., Sosa, P., R. Becchio, Arce, J., and Marelli, H.: Ground/satellite observations and atmospheric modeling of dust storms originating in the high Puna–Altiplano deserts (South America): Implications for the interpretation of paleo–climatic archives, Journal of Geophysical Research: Atmospheres, 118, 3817-3831, 2013. Hamzeh, M. A., Gharaie, M. H. M., Lahijani, H. A. K., Djamali, M., Harami, R. M., and Beni, A. N.: Holocene hydrological changes in SE Iran, a key region between Indian Summer Monsoon and Mediterranean winter precipitation zones, as revealed from a lacustrine sequence from Lake Hamoun, Quaternary International, 408, 25-39, 2015. Harman, J. R.: Synoptic Climatology of the Westerlies: Process and Patterns, Association of American Geographers, Washington, DC, 1991. Kirby, M., Patterson, W., Mullins, H., and Burnett, A.: Post-Younger Dryas climate interval linked to circumpolar vortex variability: isotopic evidence from Fayetteville Green Lake, New

York, Climate Dynamics, 19, 321-330, 2002. Li, Y., Song, Y. G., Lai, Z. P., Han, L., and An, Z. S.: Rapid and cyclic dust accumulation during MIS 2 in Central Asia inferred from loess OSL dating and grain-size analysis, Scientific Reports, 6, 2016. Liu, J. T., Jiang, X. G., Zheng, X. J., Kang, L., and Qi, F. Y.: An intensive Mongolian cyclone genesis induced severe dust storm, Terrestrial, Atmospheric and Oceanic Sciences, 15, 1019-1033, 2004. Lydolph, P. E.: Climates of the Soviet Union, World Survey of Climatology, Elsevier, Amsterdam, 1977. Machalett, B., Oches, E. A., Frechen, M., Zoller, L., Hambach, U., Mavlyanova, N. G., Markovic, S. B., and Endlicher, W.: Aeolian dust dynamics in central Asia during the Pleistocene: Driven by the long-term migration, seasonality, and permanency of the Asiatic polar front, Geochem Geophy Geosy, 9, 2008. Prins, M. A., and Vriend, M.: Glacial and interglacial eolian dust dispersal patterns across the Chinese Loess Plateau inferred from decomposed loess grain-size records, Geochem Geophy Geosy, 8, 2007. Prins, M. A., Vriend, M., Nugteren, G., Vandenberghe, J., Lu, H. Y., Zheng, H. B., and Weltje, G. J.: Late Quaternary aeolian dust input variability on the Chinese Loess Plateau: inferences from unmixing of loess grain-size records, Quaternary Sci Rev, 26, 230-242, 2007. Ramisch, A., Lockot, G., Haberzettl, T., Hartmann, K., Kuhn, G., Lehmkuhl, F., Schimpf, S., Schulte, P., Stauch, G., Wang, R., Wünnemann, B., DadaYan, Zhang, Y., and Diekmann, B.: A persistent northern boundary of Indian Summer Monsoon precipitation over Central Asia during the Holocene, Scientific reports, 6, 2016. Schöne, B. R., Oschmann, W., Tanabe, K., Dettman, D., Fiebig, J., Houk, S. D., and Kanie, Y.: Holocene seasonal environmental trends at Tokyo Bay, Japan, reconstructed from bivalve mollusk shells—implications for changes in the East Asian monsoon and latitudinal shifts of the Polar Front, Quaternary Sci Rev, 23, 1137-1150, 2004. Shriner, D. S., and Street, R. B.: North America, in The Regional Impacts of Climate Change, an Assessment of Vulnerability, edited by R. T. Watson, M. C. Zinyowera, and R. H. Moss, chap. 8, Cambridge Univ. Press, Cambridge, U. K., 1998. Vandergoes, M. J., Newnham, R. M., Preusser, F., Hendy, C. H., Lowell, T. V., Fitzsimons, S. J., Hogg, A. G., Kasper, H. U., and Schlüchter, C.: Regional insolation forcing of late Quaternary climate change in the Southern Hemisphere, Nature, 436,
242-245, 2005. Vriend, M., Prins, M. A., Buylaert, J. P., Vandenberghe, J., and Lu, H. Y.: Contrasting dust supply patterns across the north-western Chinese Loess Plateau during the last glacial-interglacial cycle, Quaternary International, 240, 167-180, 2011. Zhang, X. Y., Arimoto, R., and An, Z. S.: Glacial and interglacial patterns for Asian dust transport, Quaternary Sci Rev, 18, 811-819, 1999.

———————————————————

**Fig. 1.** Comparison of end-member unmixing results of NLK loess and Xiaoerbulake (XEBLK) loess grain-size distributions.

**Fig. 2.** (Nd/Yb)N vs. (La/Gd)N of loess, top soil and desert sands from the Ili Basin and Kazakhstan.

**500hPa 79-14 Jan average wind**

**500hPa 79-14 Jul average wind**

7.2  9.6  12  14.4  16.8  19.2  21.6  24  26.4

1.2  2.4  3.6  4.8  6  7.2  8.4  9.6  10.8

a

b

**850hPa 79-14 Jan average wind**

**850hPa 79-14 Jul average wind**

1.1  2.2  3.3  4.4  5.5  6.6  7.7  8.8  9.9

1.8  3.6  5.4  7.2  9  10.8  12.6  14.4  16.2

c

d

**Fig. 3.** Mean streamline (m/s) for the years 1979–2014 in Central Asia (data from the European Centre for Medium-Range Weather Forecasts).

---

## Author Comment (AC3) · 12 May 2017

CS: Dear authors, I agree with most of your replies and thank you for the modifications. I just want to react with 2 comments: 1. To the origin of the very fine silt-clay component: Chemical weathering is indeed a good candidate as measured by Konert and Vandenberghe 1997, and well illustrated by the experiments of Sun YB et al 2006. Transport as aggregates of fines by monsoonal dust storms (Qiang et al 2010) is contradicted by their very widespread and general occurrence (Vandenberhe 2013). Adherence of fines to larger grains has been contradicted by several authors. 2. Provenance of EM 1-2: I agree with your explanation. I understand now that you also agree with a northern wind, however not crossing the high mountains to the north but carrying dust only at low elevation over short distance. In my opinion, the carrying agent may still be the northern monsoonal wind, although restricted to the Ili basin.

AR:

1. The complexity of finer component is reflected in not only its origin but also uncertainty of instrument measurement (Ujvari et al., 2016;Mason et al., 2011). Chemical weathering can efficiently decrease gain size of loess (or paleosol) through the transformation of feldspar minerals into clay minerals linked closely to the process of pedogenesis. Sun et al. (2011) regarded this component formed by pedogenesis "ultrafine component". However, we have investigated clay mineralogy of NLK loess section, and results showed that the major clay mineral components in the NLK section were illite, chlorite, kaolinite and smectite, and that those clay minerals mainly had detrital origin, and rather than are in-situ weathered products. Moreover, variations in illite contents along the NLK section may be controlled by wind intensity, because weaker wind intensity would transport more fine fractions, which was supported by the wind tunnel experiment (Wang et al., 2017). Therefore, we think the degree of influence of chemical weathering on the loess grain size depends on the differences of environment conditions from site to site. Qiang et al. (2010) suggested that formation of aggregation increased particle mass, which enabled fine grains to be deposited even under stronger winds by dry deposition, however, the aggregates had larger pores and relevant lower density than individual minerals grain of the same size. Therefore the aggregates still can be influenced by the effects of sorting by aeolian processes. However, by observing the dust deposition collected in dust storm, Lin et al. (2016) thought that particles less than 20 $\mu$m could settle down during floating dust weather when the wind velocity decreased and even stopped. Therefore, it seemed to be difficult to distinguish that the aggregates were formed after deposition or they were transported by winds directly. Observations of modern dust under the scanning electron showed the phenomena of

aggregation and/or fine particles adhering to larger ones (Pye, 1987, 1995;Derbyshire et al., 1998;Falkovich et al., 2001;Qiang et al., 2010), whereas the micrographs of fresh samples from the southern margin of Tarim Basin under SEM showed little aggregation, or adhering of fine particles to the coarse particles (Lin et al., 2016). Maybe more convincing evidence will come from a lot of studies of modern storm processes.

2. For provenance of EM1 and EM2, it seems we cannot exclude the influence of monsoonal wind as a carrying agent from northeast. Gurbantunggut Desert in the Junggar Basin is a large source area of aeolian dust. However, "the upper limit of the loess distribution of 2400 m above sea level (asl) is much lower than the average elevation of about 4000 m of the northern Tianshan Mts. downwind" (Sun, 2002). And the Junggar Basin is not a frequent dust storm outbreak region (cf. Fig. 7 in Sun (2002)). Moreover, airmass backward trajectory was performed for April of this year using the HYSPLIT model (https://ready.arl.noaa.gov/HYSPLIT.php). Although the backward trajectory results may contain some uncertainties due to the uncertainty in meteorological data, Fig. 1 showed that an atmospheric circulation was formed in the Junggar Basin, and the northern Tianshan Mts. could serve as the southern boundary of this circulation. Therefore, we prefer to consider the surface-level air from west as the main transport agent of NLK loess, and maybe there is a tiny amount of dust as a background sedimentation reaching the Ili Basin crossing the high mountains, but their grain size are very fine.

References

Derbyshire, E., Meng, X. M., and Kemp, R. A.: Provenance, transport and characteristics of modern aeolian dust in western Gansu Province, China, and interpretation of the Quaternary loess record, J Arid Environ, 39, 497-516, DOI 10.1006/jare.1997.0369, 1998.

Falkovich, A. H., Ganor, E., Levin, Z., Formenti, P., and Rudich, Y.: Chemical and mineralogical analysis of individual mineral dust particles, J Geophys Res-Atmos, 106,

18029-18036, Doi 10.1029/2000jd900430, 2001.

Lin, Y. C., Mu, G. J., Xu, L. S., and Zhao, X.: The origin of bimodal grain-size distribution for aeolian deposits, Aeolian Res, 20, 80-88, 10.1016/j.aeolia.2015.12.001, 2016.

Mason, J. A., Greene, R. S., and Joeckel, R. M.: Laser diffraction analysis of the disintegration of aeolian sedimentary aggregates in water, Catena, 87, 107-118, 2011.

Pye, K.: Aeolian Dust and Dust Deposits, in, Academic Press, London, 29-62, 1987.

Pye, K.: The nature, origin and accumulation of loess, Quaternary Sci Rev, 14, 653-667, Doi 10.1016/0277-3791(95)00047-X, 1995.

Qiang, M., Lang, L., and Wang, Z.: Do fine-grained components of loess indicate westerlies: Insights from observations of dust storm deposits at Lenghu (Qaidam Basin, China), J Arid Environ, 74, 1232-1239, 10.1016/j.jaridenv.2010.06.002, 2010.

Sun, D., Su, R., Li, Z., and Lu, H.: The ultrafine component in Chinese loess and its variation over the past 7.6 Ma: implications for the history of pedogenesis, Sedimentology, 58, 916-935, 2011.

Sun, J. M.: Provenance of loess material and formation of loess deposits on the Chinese Loess Plateau, Earth Planet Sc Lett, 203, 845-859, Pii S0012-821x(02)00921-4 Doi 10.1016/S0012-821x(02)00921-4, 2002.

Ujvari, G., Kok, J. F., Varga, G., and Kovacs, J.: The physics of wind-blown loess: Implications for grain size proxy interpretations in Quaternary paleoclimate studies, Earth-Sci Rev, 154, 247-278, 10.1016/j.earscirev.2016.01.006, 2016.

Wang, X., Lang, L., Hua, T., Zhang, C., and Li, H.: The effects of sorting by aeolian processes on the geochemical characteristics of surface materials: a wind tunnel experiment, Frontiers of Earth Science, 1-9, 2017.
* * *
NOAA HYSPLIT MODEL
Backward trajectories ending at 0200 UTC 30 Apr 17
GFSG Meteorological Data

*Height: 100 m AGL*
*Trajectory Direction: Backward*
*Duration: 720 hrs*
*Vertical Motion Calculation Method: Model Vertical Velocity*
*Meteorology: 0000Z 30 Apr 2017 - GDAS0p5*

Elevation
7000 m
5000 m
3500 m
2000 m
1000 m
500 m
200 m
50 m
0 m

**Fig. 1.** Backward trajectory results for Tianshan area during April in 2017, showing the potential dust transport paths

---

## Referee Comment (RC3) · Anonymous Referee #2 · 25 Jul 2017

Linguistic issues: Lines 37-42: A lack of correlation between EM1 proportions and GISP $\delta$18O values at the millennial scale, combined with modern weather data, suggests that Arctic polar front predominates in the Ili Basin and the Kyrgyz Tian Shan piedmont during cold phases, which leads to the dust transport and accumulation of loess deposits, while the shift of mid-latitude westerlies towards the south and north controls the patterns of precipitation/moisture variations in this region. Reviewer's note: a lack of correlation between A and B means C was dominant? It implies that there are no other possibilities (D, E, . . .). Even worse, is "while the shift of the mid-latitude

westerlies . . ..controls patterns of precipitation/moisture . . ." corresponding to or with "shift of the Arctic polar front controls the temperature patterns of wind strength"? If so, you have to say so.

Lines 42-44: Comparison of EM1 proportions with Northern Hemisphere summer insolation clearly illustrates local insolation-based control on wind dynamics in the region, and humidity can also influence grain size of loess over MIS3 in particular. Reviewer's note: to me (this reviewer), the logic relationship between these two sentences are not traceable at all. "local insolation-based control on wind dynamics": what does this mean?

Lines 55-60: The relative influence and intensity of these major climate subsystems have varied across the latitudinal and longitudinal range of Central Asia through time. Thus identification of the predominant climate regimes in a certain region is a crucial precondition for tracing paleoclimatic evolution. Reviewer's note: (1) relative influence? Maybe relative importance. (2) The first sentence continues its SPECIFIC tone (i.e., Central Asian), but the second sentence turns to a general tone (i.e., a certain region). To me (this reviewer), it is misleading.

Lines 66-72: While loess in Central Asia has (. . ...) increasingly formed the focus of loess research, as yet the forcing mechanisms and the climatic conditions responsible for loess-paleosol sequences formation are ambiguous, and the paleoclimatic evolution recorded by these loess deposits in this region is not systematically understood. Reviewer's note: to me (this reviewer), "increasingly formed the focus", "the forcing mechanisms. . .are ambiguous", and "not systematically understood"are all belong to "expression inadequacies".

Lines 78-81: Climatic teleconnections, especially between the North Atlantic and East Asian Monsoon regions, are likely to have been recorded within the Central Asian loess. As yet, however, the region so far largely lacks data by which the role and contribution of the central parts of the Eurasian continent, as an environmental bridge,

can be elucidated. Reviewer's note: to me (this reviewer), there is a logic gap in this statement. I mean that you (authors) may have to bring the environmental bridge to the front so that the importance of Central Asia in documenting the teleconnections is pronounced first.

FURTHER NOTES: MY SPECIFIC "LINGUISTIC COMMENT" STOPS HERE AND OF COURSE THERE ARE MORE LINGUISTIC ISSUES. THIS PAPER IS READABLE, BUT THE WRITING SHOULD BE GREATLY IMPROVED.

Other suggestions Magnetic Susceptibility 1.1. "Low susceptibility in paleosols and high susceptibility in loess units" were sufficiently documented in Alaskan loess and in Siberian loess and Professor Liu Xiuming is a leading scientist on this. Please see if his works and propositions can help you. 1.2. The coarse particle-association of high susceptibility can be tested simply by measuring the susceptibility of different particle-size fractions. This can be done on selected samples and the data of the selected samples may elevate your confidence of interpretation. 1.3. If I were the author, I would have completely excluded susceptibility portion from this paper and may (just may) write a separate paper on magnetic susceptibility.

Particle Size 2.1. You need a comprehensive and streamlined review on existing literature dealing with interpretation of loess particle size. The literature review can be either "school division-based" or time-based (earlier time and later time) or country-based (west and China). 2.2. After the expected review is properly done, you may delete those insignificant references (I mean that you cited two many and that many of them may be insignificant). 2.3. Since you heavily rely on Vandenburghe (2013) for EM1, EM2, and EM3 arguments, you are strongly suggested to provide a complete and concise re-statement of Vandenburghe (2013) in debating pros and cons of EM1, EM2, and EM3 for representing aolian dynamics. If he was so sure and nobody else was at his odd, your application of EM1, EM2, and EM3 to interpreting aolian dynamics may be more acceptable. If his argument was case-dependent, you have a harder task to establish your case though. 2.4. I am wondering if the cumulative particle-size curve

does show a statistically meaningful break between EM1 and EM2 and also a break between EM3 and EM2? If it does not, should your reliance on Vandenburghe (2013) be questionable? What I try to say is: if you can confidently justify the acceptance of EM1, EM2, and EM3 for representing aolian dynamics, you do have a case here. Otherwise, your opponents can always argue that: those coarse particles may have indeed locally sourced, but those fine particles can either be remotely (high-elevation) sourced or locally (near-surface) sourced.

Questions for 5.3 Aeolian dust dynamics in eastern Central Asia: links to atmospheric systems Lines 440-447: Central Asia is variably influenced by the Asian monsoon from the south (Dettman et al., 2001; Cheng et al., 2012), the mid-latitude westerlies (Vandenberghe et al., 2006), the Siberian high-pressure systems from the northeast (Youn et al., 2014), and the polar front from the north (Machalett et al., 2008). However, by virtue of its geographical position, most of these climate influences can be excluded for the Ili Valley since it is sheltered to the northeast, east and south. The Asian high mountains largely inhibit the intrusion of Asian (Indian and East Asian) monsoons to the region, and the influence of the Siberian High (An, 2000) has been shown to decrease westward from the CLP (Vandenberghe et al., 2006). Reviewer's note: Downplaying Asian monsoons may be acceptable since the Yili Valley is indeed blocked by the Tianshan Mountains on the south. But, downplaying Siberian high-pressure system (SibH) is not well justified. Yes, SibH is weakening away from its center, but you cannot say that the Yili Valley was beyond the SibH influence. Furthermore, your favored "polar front" is actually also blocked by high mountains on the north. If polar front was indeed the major player, you may have to provide modern climate backgrounds in which strong polar front interacted with the prevailing westerly flow to stimulate dust storms in the Yili Valley.

Lines 448-456: Modern satellite data indicates that dust storm development in Ili river valley is closely linked with southward-moving high-latitude air masses (Ye et al., 2003). Karger et al. (2016) provided a detailed picture of the westerlies for the Ili Basin,

in which a rain belt gradually migrated towards the south and north in autumn and summer, respectively. According to this scenario, enhanced evaporation coupled with strengthened westerly winds would bring more humid and warm air masses to Arid Central Asia (ACA) during the Holocene (Zhang et al., 2016). Therefore, based on our grain-size observations, we argue that the Arctic polar front, intruding southward in the winter and retracting northward in summer (Machalett et al., 2008), most likely increased the frequency and strength of cyclonic storms, leading to dust transport and the accumulation of loess deposits during cold phases when it predominated in the Ili Basin and along the Kyrgyz Tian Shan piedmont. Reviewer's note: I (this reviewer) failed to see the linkage between "souward-moving high-latitude air masses" and "migrated rain belt". I also failed to see the linkage between "enhanced evaporation" and "strengthened westerly winds". Consequently, I failed to see the logic of your reasoning: the Arctic polar front, intruding southward in the winter and retracting northward in summer (Machalett et al., 2008), most likely increased the frequency and strength of cyclonic storms during cold phases. At least, you have to say more about the logic of your reasoning.

---

## Editor Comment (EC1) · L. Zhou (Editor) · 24 Aug 2017

May I ask you reply to the comments by Referee 2?

---

## Author Comment (AC4) · 29 Aug 2017

Linguistic issues:

CS: Lines 37-42: A lack of correlation between EM1 proportions and GISP $\delta$18O values at the millennial scale, combined with modern weather data, suggests that Arctic polar front predominates in the Ili Basin and the Kyrgyz Tian Shan piedmont during cold phases, which leads to the dust transport and accumulation of loess deposits, while the shift of mid-latitude westerlies towards the south and north controls the patterns of

precipitation/moisture variations in this region. Reviewer's note: a lack of correlation between A and B means C was dominant? It implies that there are no other possibilities (D, E, . . .). Even worse, is "while the shift of the mid-latitude westerlies . . . controls patterns of precipitation/moisture . . ." corresponding to or with"shift of the Arctic polar front controls the temperature patterns of wind strength"? If so, you have to say so.

AR: Thanks for your critical comments. The logic of the abstract was unclear, and it is unreasonable to draw conclusions beyond the information available in the data. We have now tried to clear the confused logic, and rewritten the Abstract and Conclusions sections in this manuscript. It is important to note that Central Asia is very large and consequently it is reasonable to assume that different climate subsystems act upon different parts of the region. Therefore, observations made at one end of Central Asia (e.g. Tajikistan) do not necessarily apply to the other (e.g. Ili Basin). Furthermore, the Ili Basin itself is almost 1000km across and is geographically diverse, and it is reasonable to assume that the western part of the basin, e.g. the published site of Remizovka, is more exposed to influences such as the polar northerlies than sites in the eastern part of the basin, e.g. NLK presented here, which are much more sheltered by the high Tien Shan mountains. Tajikistan is mainly impacted by the westerlies, and the North Atlantic climatic signals are presented in Tajikistan loess, which implies that the westerlies linking the North Atlantic and the Eurasia loess, can influence accumulation of loess deposits in Tajikistan (Vandenberghe et al., 2006). A lack of correlation between EM1 proportions and GISP $\delta$18O values at the millennial scale only indicates that other climate systems control the wind dynamics responsible for dust transport and the accumulation of loess during cold phases in NLK, rather than the Westerlies. Thus, we cannot conclude that "Arctic polar front predominates in the Ili Basin and the Kyrgyz Tian Shan piedmont during cold phases." We have rewritten this sentence, like this "The dominance of the Siberian high-pressure system at NLK in eastern Central Asia contrasts with the influence of the mid-latitude westerlies in southwest Central Asia, as demonstrated by good correspondence between grain size and GISP $\delta$18O at Darai Kalon, Tajikistan." We have deleted the unreasonable suggestion that the shift

of mid-latitude westerlies towards the south and north controls the patterns of precipitation/moisture variations in Central Asia. We added some records from modern and Holocene climate change records to substantiate our arguments for mid-Westerlies changes. Actually, those records demonstrated that the mid-latitude Westerlies truly controlled the patterns of moisture variations in Arid Central Asia (ACA) (Li et al., 2011; Huang et al., 2015; Cai et al., 2017). However, we can't draw that conclusion from "A lack of correlation between EM1 proportions and GISP $\delta$18O values at the millennial scale". "while the shift of the mid-latitude westerlies ... controls patterns of precipitation/moisture ..." isn't corresponding to or with "shift of the Arctic polar front controls the temperature patterns of wind strength". Available data enable us to compare our data from the eastern, sheltered end of the Ili Basin with the more exposed Remizovka section at the southwestern margins of the basin – with respect to likely climatic influence and its impact on grain size. Remisowka (Machalett et al., 2008) is located along the northern piedmont of Tianshan Mountains (Fig. 1a in manuscript). Because NLK is much more sheltered from northerly weather systems than Remisowka, there is a good chance that the polar front had more of an influence on Remisowka than on NLK. In addition, based on modern and Holocene climate data, we argue that the Siberian High may have exerted a significant influence on wind dynamics in the Ili Basin, leading to dust transport and the accumulation of loess during cold phases in NLK. Thus we argue that the Siberian High controls wind strength and mid-latitude westerlies control precipitation/moisture. A strengthened Siberian High would push the mid-latitude Westerlies pathways further to the south, resulting in comparably drier conditions in the northern Central Asia regions (e.g. Tianshan Mountains) but wetter conditions in south-western Central Asia (Pamir) (Lei et al., 2014; Wolff et al., 2017). Intensity and geographical position of the Siberian High can strongly control precipitation and atmospheric circulation patterns (meridional or zonal) at mid-latitudes of Asia (Panagiotopoulos et al., 2005). The coupling of the Siberian High with the mid-latitude Westerlies system likely contributed significantly to the climate variability in the study area. We have modified our text to explain these drivers more clearly.

CS: Lines 42-44: Comparison of EM1 proportions with Northern Hemisphere summer insolation clearly illustrates local insolation-based control on wind dynamics in the region, and humidity can also influence grain size of loess over MIS3 in particular. Reviewer's note: to me (this reviewer), the logic relationship between these two sentences are not traceable at all. "local insolation-based control on wind dynamics": what does this mean?

AR: We have clarified our expression. In Fig. 7, we can see the consistent variation trend between EM1 proportions and summer insolation, which illustrated the great influences of insolation on the grain size, i.e. wind dynamics. EM1 proportions appear to correlate well with Northern Hemisphere summer insolation. We used June insolation at 45°N in this paper since the latitude is similar to that of NLK section. Thus we changed the term "local insolation" to "summer insolation at 45°N" / "regional insolation variation" in the manuscript in order to avoid misunderstanding. The grain-size variations over MIS 3 don't correspond to the trend of summer insolation, however. Since moisture availability is the dominant factor controlling glacier growth in Central Asia, we attributed that to influences of humidity on vegetation growth in source areas and further availability of source sediment. Consequently, we rewrote this sentence, like this "On orbital timescales, comparison of EM1 proportions with Northern Hemisphere summer insolation at 45°N strongly suggests control of summer insolation on wind dynamics in the region. Out of phase peaks in loess deposition and glacial expansion in the region demonstrate the importance of moisture availability on loess deposition in MIS3 in particular."

CS: Lines 55-60: The relative influence and intensity of these major climate subsystems have varied across the latitudinal and longitudinal range of Central Asia through time. Thus identification of the predominant climate regimes in a certain region is a crucial precondition for tracing paleoclimatic evolution. Reviewer's note: (1) relative influence? Maybe relative importance. (2) The first sentence continues its SPECIFIC tone (i.e., Central Asian), but the second sentence turns to a general tone (i.e., a certain region). To me (this reviewer), it is misleading.

AR: (1) We have clarified this distinction in the text, and substituted "relative influence" with "relative importance". (2) We cannot use a specific concept to represent a general concept. It is indeed misleading. We have changed the second sentence to "Thus identification of the predominant climate regimes in this region, using geological archives, is a crucial precondition for tracing paleoclimatic evolution."

CS: Lines 66-72: While loess in Central Asia has (. . .. . .) increasingly formed the focus of loess research, as yet the forcing mechanisms and the climatic conditions responsible for loess-paleosol sequences formation are ambiguous, and the paleoclimatic evolution recorded by these loess deposits in this region is not systematically understood. Reviewer's note: to me (this reviewer), "increasingly formed the focus", "the forcing mechanisms . . . are ambiguous", and "not systematically understood" are all belong to"expression inadequacies".

AR: Here, we have simplified the language and made the purpose of this paper much clearer and better to understand. We also added three citations in an effort to reinforce the lack of systematic understanding of the forcing mechanisms and the climatic conditions responsible for loess-paleosol sequences formation.

CS: Lines 78-81: Climatic teleconnections, especially between the North Atlantic and East Asian Monsoon regions, are likely to have been recorded within the Central Asian loess. As yet, however, the region so far largely lacks data by which the role and contribution of the central parts of the Eurasian continent, as an environmental bridge, can be elucidated. Reviewer's note: to me (this reviewer), there is a logic gap in this statement. I mean that you (authors) may have to bring the environmental bridge to the front so that the importance of Central Asia in documenting the teleconnections is pronounced first.

AR: Thanks for your suggestion. We have clarified the language in the text and the wording of our arguments. Since we know basically nothing about millennial-scale

climatic changes in Central Asia, our aim is to investigate a loess section in Central Asia to see to what degree climatic teleconnections exist between North Atlantic and East Asia first, i.e. the first step is to generate data. Therefore, we have made the aim clearer, like this "Data for Central Asian loess are so far lacking at this resolution, despite its strategic location as a likely environmental bridge between the North Atlantic and East Asian Monsoon regions." We deleted the sentence "Climatic teleconnections, especially between the North Atlantic and East Asian Monsoon regions, are likely to have been recorded within the Central Asian loess."

CS: Other suggestions Magnetic Susceptibility 1.1. "Low susceptibility in paleosols and high susceptibility in loess units" were sufficiently documented in Alaskan loess and in Siberian loess and Professor Liu Xiuming is a leading scientist on this. Please see if his works and propositions can help you. 1.2. The coarse particle-association of high susceptibility can be tested simply by measuring the susceptibility of different particle size fractions. This can be done on selected samples and the data of the selected samples may elevate your confidence of interpretation. 1.3. If I were the author, I would have completely excluded susceptibility portion from this paper and may (just may) write a separate paper on magnetic susceptibility.

AR: The relationship between pedogenesis and magnetic susceptibility in the higher-latitude loess deposits of Alaska and Siberia is different from the Chinese Loess Plateau loess as suggested by Liu et al. (1999) and Liu et al. (2008). At NLK, lower susceptibility exists in paleosols and higher susceptibility in loess units. Although this scenario is difficult to explain fully through variation in wind strength alone, it showed that wind strength, or wind dynamics, would influence MS variations at least and thus paleoclimatic reconstruction using climatic proxies, such as MS. Thus it is necessary to understand the atmospheric dynamic pattern during loess deposition further. Consistently low $\chi$fdUnderstanding the mechanisms for the enhancement of magnetic susceptibility is beyond the scope of this study. We only intended to illustrate the significant impacts of wind dynamic on MS. In addition, ferromagnetic minerals, including

magnetite and hematite, belong to heavy minerals which have higher relative density. Thus when wind becomes stronger, more ferromagnetic minerals will be transported to deposition areas, resulting to higher MS values. Thus we have modified the subtitle 5.1, like this "Impacts of wind dynamics on magnetic susceptibility variations".

CS: Particle Size 2.1. You need a comprehensive and streamlined review on existing literature dealing with interpretation of loess particle size. The literature review can be either "school division-based" or time-based (earlier time and later time) or country based (west and China). 2.2. After the expected review is properly done, you may delete those insignificant references (I mean that you cited too many and that many of them may be insignificant). 2.3. Since you heavily rely on Vandenburghe (2013) for EM1, EM2, and EM3 arguments, you are strongly suggested to provide a complete and concise re-statement of Vandenburghe (2013) in debating pros and cons of EM1, EM2, and EM3 for representing aolian dynamics. If he was so sure and nobody else was at his odd, your application of EM1, EM2, and EM3 to interpreting aolian dynamics may be more acceptable. If his argument was case-dependent, you have a harder task to establish your case though. 2.4. I am wondering if the cumulative particle-size curve does show a statistically meaningful break between EM1 and EM2 and also a break between EM3 and EM2? If it does not, should your reliance on Vandenburghe (2013) be questionable? What I try to say is: if you can confidently justify the acceptance of EM1, EM2, and EM3 for representing aolian dynamics, you do have a case here. Otherwise, your opponents can always argue that: those coarse particles may have indeed locally sourced, but those fine particles can either be remotely (high-elevation) sourced or locally (near-surface) sourced.

AR: Thanks for your suggestions. In the section 5.2, we have summarized the significance of grain-size analysis. Relevant studies were separated into two groups according to the unmixing method of grain size spectra. Vandenberghe (2013) applied visual inspection of grain-size distribution curves and the EMMA end-member analysis in combination to define the characteristic grain-size distribution of primary loess

deposits and review their respective processes and conditions of transports and deposition, relying largely on loess samples from central and eastern Asia and northwestern and central Europe. Thus his argument was based on a large number of previous studies from a range of sites, and is not case-dependent. For example, the subgroup 1.b.2 in Vandenberghe (2013) has also been identified in the loess of Chinese Loess Plateau, southern, northwestern and central Europe. Furthermore, in the studies of loess sediments from the Qilian Mountain region, Rasmussen et al. (2014), Nottebaum et al. (2015) and Yang et al. (2016) have interpreted the multiple sources of loess sediments and dynamic conditions according to sediment groups in Vandenberghe (2013). We have included those arguments in the main text. EM1 and EM2 of our results have modal grain size approximately corresponding to the 'subgroup 1.b.1' and 'subgroup 1.b.2' respectively. Vandenberghe (2013) suggested that although component 1.b.1 and 1.b.2 occur jointly together in the proximal depositional regions, they are clearly distinct from each other in terms of the coverage and transportation distance. In Fig. 4 of manuscript, the mirror image relationships over millennial scales can be observed, which may implied that both EM1 and EM2 have a same origin, but wind strength controlled the relative proportions of both through time. In addition, grain-size distributions of modern dust illustrate a modal grain size of 33.3 $\mu$m in winter and 44.6 $\mu$m in summer in the northern and western Chinese Loess Plateau (Sun et al., 2003) (Fig. 1). These modes are similar to EM2 and EM1 in our results, respectively. It is generally assumed that vegetation coverage is more extensive in summer than in winter in CLP. Therefore, availability of sediments in source areas wouldn't influence the grain sizes, conversely differences in wind dynamic between these two seasons likely play an important role in controlling the grain sizes. While EM3 ("subgroup 1.c.1") indicated a different aerodynamic environment from EM2. The former would settle when the wind velocity decreases and even stops, as suggested by Lin et al. (2016), but the latter were interpreted as transportation during cyclonal dust storm outbreaks (Vandenberghe, 2013). Consequently, the cumulative particle-size curve can give a statistically meaningful break between EM1 and EM2 and also a break between EM3

and EM2. Actually, greater dispute exists in the origin of the EM3 size fraction. In the manuscript, we suggest that the fine-grained EM3 (c. 18.9 $\mu$m) is the result of background loess supply in the Ili Basin, and infer the EM3 modal peak to derive from low altitude non-dust storm processes after excluding the aggregate model, transportation by high-altitude westerlies and influences of post-depositional processes. Therefore, those fine particles are also likely to be locally (near-surface) sourced.

CS: Questions for 5.3 Aeolian dust dynamics in eastern Central Asia: links to atmospheric systems Lines 440-447: Central Asia is variably influenced by the Asian monsoon from the south (Dettman et al., 2001; Cheng et al., 2012), the mid-latitude westerlies (Vandenberghe et al., 2006), the Siberian high-pressure systems from the northeast (Youn et al., 2014), and the polar front from the north (Machalett et al., 2008). However, by virtue of its geographical position, most of these climate influences can be excluded for the Ili Valley since it is sheltered to the northeast, east and south. The Asian high mountains largely inhibit the intrusion of Asian (Indian and East Asian) monsoons to the region, and the influence of the Siberian High (An, 2000) has been shown to decrease westward from the CLP (Vandenberghe et al., 2006). Reviewer's note: Downplaying Asian monsoons may be acceptable since the Yili Valley is indeed blocked by the Tianshan Mountains on the south. But, downplaying Siberian high-pressure system (SibH) is not well justified. Yes, SibH is weakening away from its center, but you cannot say that the Yili Valley was beyond the SibH influence. Furthermore, your favored "polar front" is actually also blocked by high mountains on the north. If polar front was indeed the major player, you may have to provide modern climate backgrounds in which strong polar front interacted with the prevailing westerly flow to stimulate dust storms in the Yili Valley.

AR: Thanks for your good suggestions. We have collected some modern and Holocene records about atmospheric circulation in central Asia over the past month. Available data enable us to compare our data from the eastern, sheltered end of the Ili Basin with the Remizovka section at the southwestern margins of the basin – with respect

to likely climatic influence and its impact on grain size. Remisowka (Machalett et al., 2008) is located along the northern piedmont of Tianshan Mountains (Fig. 1a in manuscript). Because NLK site is much more sheltered from northerly weather systems than Remisowka, there is a good chance that the polar front had more of an influence on Remisowka than on NLK. While in the north/northeast of our study area is a massive cold high — Siberian High. The Siberian High is the most dominant Northern Hemisphere anticyclone and is centered between 40°N and 65°N, 80°E and 120°E (cf. Fig. 3 in Huang et al. (2011)), and its anticyclonic feature is broadly recognized as the dominant mode of winter and spring climate over Eurasia (Sahsamanoglou et al., 1991; Savelieva et al., 2000; Gong and Ho, 2002; Panagiotopoulos et al., 2005). In addition, based on modern and Holocene climate data, we argue that the Siberian High may have exerted a significant influence on wind dynamics in the Ili Basin, leading to dust transport and the accumulation of loess during cold phases in NLK. In addition, modern meteorological data show that the maximum wind at NLK mainly blows from the west, and that dust storm development in Ili river valley is closely linked with southward-moving high-latitude air masses, while the air masses can enter into the Ili Basin round the northern Tianshan (see the Supplementary materials and Ye et al. (2003)). Therefore, the Siberian high-pressure system is able to influence the Ili Basin, and the southward-moving high-latitude air masses associated with it can enter into the Ili Basin, leading to dust transport and the accumulation of loess deposits during cold phases in NLK. We have rewritten the section 5.3. Please see Lines 449-475 in the manuscript and Supplementary materials.

CS: Lines 448-456: Modern satellite data indicates that dust storm development in Ili river valley is closely linked with southward-moving high-latitude air masses (Ye et al., 2003). Karger et al. (2016) provided a detailed picture of the westerlies for the Ili Basin, in which a rain belt gradually migrated towards the south and north in autumn and summer, respectively. According to this scenario, enhanced evaporation coupled with strengthened westerly winds would bring more humid and warm air masses to Arid Central Asia (ACA) during the Holocene (Zhang et al., 2016). Therefore, based

on our grain-size observations, we argue that the Arctic polar front, intruding south-ward in the winter and retracting northward in summer (Machalett et al., 2008), most likely increased the frequency and strength of cyclonic storms, leading to dust trans-port and the accumulation of loess deposits during cold phases when it predominated in the Ili Basin and along the Kyrgyz Tian Shan piedmont. Reviewer's note: I (this reviewer) failed to see the linkage between "souward-moving high-latitude air masses" and "migrated rain belt". I also failed to see the linkage between "enhanced evapo-ration" and "strengthened westerly winds". Consequently, I failed to see the logic of your reasoning: the Arctic polar front, intruding southward in the winter and retracting northward in summer (Machalett et al., 2008), most likely increased the frequency and strength of cyclonic storms during cold phases. At least, you have to say more about the logic of your reasoning.

AR: In this respect our logic was flawed. We have clarified the logic of our arguments in the text. As mentioned above, it is unreasonable to draw conclusions beyond the information available in the data. Therefore, we reconsidered the atmospheric sys-tem responsible for aeolian dust dynamics in our study area, and then rewrote and rearranged the paragraphs. As explained above, the Siberian high-pressure systems exerted a significant influence on wind dynamics responsible for dust transport and the accumulation of loess deposits during cold phases in NLK, and the mid-latitude West-erlies controlled the patterns of moisture variations in Arid Central Asia (ACA), based on modern and Holocene climate data. A strengthened Siberian High would push the mid-latitude Westerlies pathways further to the south, which resulted in comparably drier conditions in the northern Central Asia regions (e.g. Tianshan Mountains) but wetter conditions in south-western Central Asia (Pamir) (Lei et al., 2014; Wolff et al., 2017). Intensity and geographical position of the Siberian High can strongly control precipitation and atmospheric circulation patterns (meridional or zonal) at mid-latitudes of Asia (Panagiotopoulos et al., 2005). Therefore, the coupling of the Siberian High with the mid-latitude Westerlies system likely contributed significantly to the climate variability in the study area, which may interpret the seesaw relationship during MIS3

shown in Fig. 7 of manuscript.

References Cai, Y. J., Chiang, J. C. H., Breitenbach, S. F. M., Tan, L. C., Cheng, H., Edwards, R. L., and An, Z. S.: Holocene moisture changes in western China, Central Asia, inferred from stalagmites, Quaternary Sci Rev, 158, 15-28, 2017.

Gong, D. Y., and Ho, C. H.: The Siberian High and climate change over middle to high latitude Asia, Theor Appl Climatol, 72, 1-9, 2002.

Huang, W., Chen, J. H., Zhang, X. J., Feng, S., and Chen, F. H.: Definition of the core zone of the "westerlies-dominated climatic regime", and its controlling factors during the instrumental period, Science China-Earth Sciences, 58, 676-684, 10.1007/s11430-015-5057-y, 2015.

Huang, X. T., Oberhansli, H., von Suchodoletz, H., and Sorrel, P.: Dust deposition in the Aral Sea: implications for changes in atmospheric circulation in central Asia during the past 2000 years, Quaternary Sci Rev, 30, 3661-3674, 2011.

Lei, Y. B., Tian, L. D., Bird, B. W., Hou, J. Z., Ding, L., Oimahmadov, I., and Gadoev, M.: A 2540-year record of moisture variations derived from lacustrine sediment (Sasikul Lake) on the Pamir Plateau, Holocene, 24, 761-770, 2014.

Li, X., Zhao, K., Dodson, J., and Zhou, X.: Moisture dynamics in central Asia for the last 15 kyr: new evidence from Yili Valley, Xinjiang, NW China, Quaternary Sci Rev, 30, 3457-3466, 2011.

Lin, Y. C., Mu, G. J., Xu, L. S., and Zhao, X.: The origin of bimodal grain-size distribution for aeolian deposits, Aeolian Research, 20, 80-88, 10.1016/j.aeolia.2015.12.001, 2016.

Liu, X. M., Hesse, P., Rolph, T., and Beget, J. E.: Properties of magnetic mineralogy of Alaskan loess: evidence for pedogenesis, Quatern Int, 62, 93-102, 1999.

Liu, X. M., Liu, T. S., Paul, H., Xia, D. S., Jiri, C. C., and Wang, G.: Two pedogenic models for paleoclimatic records of magnetic susceptibility from Chinese and Siberian

loess, Sci China Ser D, 51, 284-293, 2008.

Machalett, B., Oches, E. A., Frechen, M., Zoller, L., Hambach, U., Mavlyanova, N. G., Markovic, S. B., and Endlicher, W.: Aeolian dust dynamics in central Asia during the Pleistocene: Driven by the long-term migration, seasonality, and permanency of the Asiatic polar front, Geochem Geophy Geosy, 9, Artn Q08q09 10.1029/2007gc001938, 2008.

Nottebaum, V., Stauch, G., Hartmann, K., Zhang, J. R., and Lehmkuhl, F.: Unmixed loess grain size populations along the northern Qilian Shan (China): Relationships between geomorphologic, sedimentologic and climatic controls, Quatern Int, 372, 151-166, 10.1016/j.quaint.2014.12.071, 2015.

Panagiotopoulos, F., Shahgedanova, M., Hannachi, A., and Stephenson, D. B.: Observed trends and teleconnections of the Siberian high: A recently declining center of action, J Climate, 18, 1411-1422, 2005.

Rasmussen, S. O., Bigler, M., Blockley, S. P., Blunier, T., Buchardt, S. L., Clausen, H. B., Cvijanovic, I., Dahl-Jensen, D., Johnsen, S. J., Fischer, H., Gkinis, V., Guillevic, M., Hoek, W. Z., Lowe, J. J., Pedro, J. B., Popp, T., Seierstad, I. K., Steffensen, J. P., Svensson, A. M., Vallelonga, P., Vinther, B. M., Walker, M. J. C., Wheatley, J. J., and Winstrup, M.: A stratigraphic framework for abrupt climatic changes during the Last Glacial period based on three synchronized Greenland ice-core records: refining and extending the INTIMATE event stratigraphy, Quaternary Sci Rev, 106, 14-28, 10.1016/j.quascirev.2014.09.007, 2014.

Sahsamanoglou, H. S., Makrogiannis, T. J., and Kallimopoulos, P. P.: Some Aspects of the Basic Characteristics of the Siberian Anticyclone, Int J Climatol, 11, 827-839, 1991.

Savelieva, N. I., Semiletov, I. P., Vasilevskaya, L. N., and Pugach, S. P.: A climate shift in seasonal values of meteorological and hydrological parameters for Northeastern

Asia, Prog Oceanogr, 47, 279-297, 2000.

Sun, D. H., Chen, F. H., Bloemendal, J., and Su, R. X.: Seasonal variability of modern dust over the Loess Plateau of China, J Geophys Res-Atmos, 108, Artn 4665 10.1029/2003jd003382, 2003.

Vandenberghe, J., Renssen, H., van Huissteden, K., Nugteren, G., Konert, M., Lu, H. Y., Dodonov, A., and Buylaert, J. P.: Penetration of Atlantic westerly winds into Central and East Asia, Quaternary Sci Rev, 25, 2380-2389, 10.1016/j.quascirev.2006.02.017, 2006.

Vandenberghe, J.: Grain size of fine-grained windblown sediment: A powerful proxy for process identification, Earth-Science Reviews, 121, 18-30, 10.1016/j.earscirev.2013.03.001, 2013.

Wolff, C., Plessen, B., Dudashvilli, A. S., Breitenbach, S. F., Cheng, H., Edwards, L. R., and Strecker, M. R.: Precipitation evolution of Central Asia during the last 5000 years, Holocene, 27, 142-154, 2017.

Yang, F., Zhang, G. L., Yang, F., and Yang, R. M.: Pedogenetic interpretations of particle-size distribution curves for an alpine environment, Geoderma, 282, 9-15, 2016.

Ye, W., Sang, C., and Zhao, X.: Spatial-temporal distribution of loess and source of dust in Xinjiang, Journal of Desert Research, 23, 514-520 (in Chinese), 2003.

Please also note the supplement to this comment:
https://www.clim-past-discuss.net/cp-2017-50/cp-2017-50-AC4-supplement.pdf

**Fig. 1.** Grain-size distributions of seasonal dusts in the northern and western CLP (Huanxian)

**Supplement:**

**Supplement**

Magnetic susceptibility values ($10^{-8}m^3kg^{-1}$) of different grain-size fractions

| Sample No. | Grain-size Fractions | | | | |
| --- | --- | --- | --- | --- | --- |
| | > 63 μm | 63-40 μm | 40-32 μm | 32-20 μm | <20 μm |
| NLK400 | 72 | 66 | 86 | 88 | **98** |
| NLK800 | 67 | 61 | 80 | **87** | 85 |
| NLK1108 | 73 | 78 | 104 | 107 | **108** |
| NLK1110 | 75 | 68 | 96 | **109** | 105 |
| NLK1400 | 79 | 83 | 94 | 100 | **111** |

Note: Red font represents maximum values in different grain-size fractions.

---

## Author Response (AR1)

**Explaining and replies to the comments and suggestions**

Ms. Ref. No.:  cp-2017-50

Title: Environmental dynamics since the last glacial in arid Central Asia: evidence from grain size distribution and magnetic properties of loess from the Ili Valley, western China

Following parts are our explanations what changes to the manuscript have made and how our replies (Answer or Reply, A&R, Black) to the reviewer's comments (comments and suggestions, C&S, Blue).

*Replies to the comments of Dr. Vandenberghe*

For *General Comments*

**C&S**: Many studies deal with the correspondence between loess deposits and climate circulation but the causal relations are indeed poorly understood. Previously, the attention has been drawn to competing circulation systems of westerlies and monsoons. Thus, relations are indeed best studied in a region where different systems could have been present at times. Consequently, the studied region is favorably situated for this timely research. A most interesting result is the importance of precipitation on the loess depositional signal. Until now most attention has been paid to temperature variability that of course also impacts the wind circulation. It seems by this study that resulting oscillations in loess deposition show a complex pattern. And this may be the reason for the fact that the oscillating loess signals in C. Asia and the NE Tibetan Plateau are not that simply correlated with D-O-events or H-events. The paper is well written, designed and archived.

**A&R**: Thanks for your positive comments. This paper made aims to investigate the causal relations between loess deposits and climate circulation. Recently, many researchers have paid much attention to paleoclimate reconstruction in transitional regions where different atmospheric systems predominate at different times, e.g. the Central Balkans (Ramisch et al., 2016), the Qinghai Lake region (Cheng et al., 2012), southwest Asia (Hamzeh et al., 2015). Our study area is likewise situated in the zone influenced by different climatic systems. It therefore becomes important prerequisite to clarify the paleoenvironmental significance of various proxies, in particular which proxies reflect temperature, which represent precipitation, and which indicate wind strength. Our manuscript explores the potential impacts of precipitation on the loess depositional signal, and identifies areas where more work needs to be done in the future.

We agree that some scientists have focused on the influence of temperature variability on wind circulation, and in particular the role of local insolation minima in driving an early onset of the LGM in the Southern Hemisphere (Vandergoes et al., 2005). We reconsider the reasons for variations in EM1 proportions, and suggest that the availabilities of source sediments which are likely impacted by development of permafrost and vegetation growth in dust source areas, are responsible for the weaker EM1 proportion fluctuations in LGM and early-MIS3. Please see the details in line 464-470 and 504-509 of revised manuscript. The aeolian loess sediments in Central Asia are more likely to respond to a complex mixture of global signals with local insolation, glacial activity and local weathering. This overlap will weaken the global signals as preserved within the loess.

For *Minor Comments*

**C&S**: L 343-359: Two different explanations are claimed for the origin of the fine-grained endmember 3 (c. 18.9 µm). The main difference seems to be, if I understand well, that one hypothesis invokes high-suspension transport while in the other one surface winds are involved. However, both hypotheses interpret that this component is the result of background loess supply (as confirmed in lines 389-395) as previously demonstrated by Prins et al (2007), Vriend et al (2011) and Zhang et al (1999). It is not realistic to separate the grain-size fractions of 2-8 µm (transported by westerlies) and 8-15 µm (=EM3, transported at low altitude) as the authors seem to do. Both components react jointly constituting background loess supplied by westerlies as described by e.g. Prins et al. (2007).

**A&R**: We agree with the reviewer on this point. The origins of the EM3 size fraction is indeed complex. For example, based on modern dust monitoring from the high-altitude subtropical Puna-Altiplano Plateau in South America, Gaiero et al. (2013) found that "Finer mode dust is deposited during event periods, which point to a dominant long-range transport, contrasting with a dominance of coarser mode observed for non-dust sampling periods, pointing to dominant local sources." Prins and Vriend (2007) and Prins et al. (2007) suggested that the clayed loess component represented the fine dust component supplied over the entire Loess Plateau by long-term suspension processes, and the high-level subtropical jet stream (westerly winds) might, at least partly, be responsible for the input of this fine-grained loess component. End-member unmixing results of Xiaoerbulake (XEBLK) loess (Li et al., 2016b) grain-size distributions show the similar EM3 component to NLK loess (Fig. R1). XEBLK loess section is also located in the Ili Baisn. That implies that the fine-grained EM3 (c. 18.9 µm) is the result of background loess supply in the Ili Basin regardless of its origin (Vriend et al., 2011;Zhang et al., 1999;Prins et al., 2007). It is difficult to determine the origins of the fine silt/clay. The appearance of the fine component in dust deposition may be caused by aggregation, due to fine particles adhering to the coarse particles, as well as chemical weathering. Perhaps the method of Machalett et al. (2008) is the better alternative. They neither removed organic matter and carbonates from the stratigraphic samples and nor applied an intensive ultrasonic treatment to disaggregate particles.

[Figure]

Fig. R1 Comparison of end-member unmixing results of NLK loess and Xiaoerbulake (XEBLK)

loess grain-size distributions.

**C&S**: L 441-447: If the Ili valley is sheltered from northeastern wind, as the authors claim, what is then the source area for the EM1 and EM2 fractions? There is no apparent difference between these coarse-grained fractions on the CLP, N Tibet Plateau and in the Ili valley where a distinct supply is clear from the northeast under the influence of the Siberian High.

**A&R**: Thank you for this suggestion. The northern Tien Shan Range reaches altitudes of > 4000 m a.s.l. For the particles with grain size of > 20 µm, it is unlikely that grains of this coarser silt fraction were transported by north-easterly winds above the 4000 m altitude over the northern Tien Shan and into the Ili Basin. We therefore interpret the coarser grained loess particles in the Ili Basin to have been predominantly transported by near-surface winds. The topographic context (Fig. 1 in revised manuscript) most likely ensured the westerly winds coming to be the transporting agent. Moreover, we added modern meteorological data (2009-2013) in NLK in the *Supplementary file*. It was evident that the strongest winds at NLK site mainly blowed from the west.

In our speculation as to the provenance of the NLK loess, we initially compared the REE parameters of NLK loess with those of desert sands and modern soils from the Ili Basin and further west into Kazakhstan (Fig. R2). Our results indicated that the deserts and topsoils in Kazakhstan are unlikely to be the main potential source areas. In contrast, topsoils from the Ili Basin probably provide the most important source materials in the NLK loess. The Quaternary sediments of the Ili Basin mainly consist of alluvial fans and floodplains, and the top soils developed on those. We therefore speculate a proximal source for the NLK loess. Furthermore, recent work from our group indicates that size-differentiated rare earth elements (REE) may help to distinguish potential proximal or distal sources (Chen et al., 2017). In future, we expect to find more substantial evidence for tracing loess provenance in the region.

[Figure]

Fig. R2 (Nd/Yb)$_N$ vs. (La/Gd)$_N$ of loess, top soil and desert sands from the Ili Basin and Kazakhstan.

**C&S**: L 454-456: Explain better how the 'cyclonic storms' originated by protrusion of the Arctic polar front, rather than by other circulation patterns.

**A&R**: This is a constructive question. We have reconsidered this issue. We have collected some modern and Holocene records about atmospheric circulation in Central Asia over the past month, and exclude the influences of the Arctic polar front and assure the importance of the Siberian High for dust transport and increased loess accumulation at NLK.

Available data enable us to compare our data from the eastern, sheltered end of the Ili Basin with the Remizovka section at the southwestern margins of the basin – with respect to likely climatic influence and its impact on grain size. Remisowka (Machalett et al., 2008) is located along the northern piedmont of Tianshan Mountains (Fig. 1a in manuscript). Because NLK site is much more sheltered from northerly weather systems than Remisowka, there is a good chance that the polar front had more of an influence on Remisowka than on NLK. While in the north/northeast of our study area is a massive cold high ─ Siberian High. The Siberian High is the most dominant Northern Hemisphere anticyclone and is centered between 40°N and 65°N, 80°E and 120°E (cf. Fig. 3 in Huang et al. (2011)), and its anticyclonic feature is broadly recognized as the dominant mode of winter and spring climate over Eurasia (Sahsamanoglou et al., 1991;Savelieva et al., 2000;Panagiotopoulos et al., 2005;Gong and Ho, 2002). In addition, based on modern and Holocene climate data, we argue that the Siberian High may have exerted a significant influence on wind dynamics in the Ili Basin, leading to dust transport and the accumulation of loess during cold phases in NLK.

In addition, modern meteorological data show that the maximum wind at NLK mainly blows from the west, and that dust storm development in Ili river valley is closely linked with southward-moving high-latitude air masses, while the air masses can enter into the Ili Basin round the northern Tianshan (see the Fig. S5 in *Supplementary file* and Ye et al. (2003)). Therefore, the Siberian high-pressure system is able to influence the Ili Basin, and the southward-moving high-latitude air masses associated with it can enter into the Ili Basin, leading to dust transport and the accumulation of loess deposits during cold phases in NLK.

Also, we compare secular trends between the EM1 proportions and mean grain size data from the Jingyuan section over the last glacial period (Sun et al., 2010). It is widely acepted that increases in grain-size records from the CLP are linked to a strengthening of the East Asian winter monsoon due to an intensification of the Siberian High (Hao et al., 2012;Ding et al., 1995). The similarities in the trends can be observed (Fig. 7 in revised manuscript). For example, there remain coarser grain size and higher sedimentation rate during mid-MIS3 (Sun et al., 2010), and opposite cases occur in early- and late-MIS3. Therefore, that supports that a common Eurasian atmospheric forcing pattern — the Siberian High — is responsible for the climate evolution of these two regions during that time period. Therefore, we have rewritten the section 5.3. Please see *Lines 439-487* in the revised manuscript.

**C&S**: L 476: The interesting absence of correlation between the observed grain-size signals and N Atlantic abrupt events is not only found in the Ili valley but also previously in Tadjikistan and the NE Tibet Plateau (Vandenberghe et al. 2006)

**A&R**: That point has been attracting the attention of our group recently. We find that EM1 proportions fluctuate weaker in H2 and H5 events (Fig. 7 in the revised manuscript). We thus reconsider the reason for variations in EM1 proportions, and suggest that the availabilities of source sediments which are likely impacted by development of permafrost and vegetation growth in dust source areas, are responsible for the smaller EM1 proportions in LGM and early-MIS3. Please see the details in line 464-470 and 504-509 in the revised manuscript.

Therefore, in our view, the lack of good correlation between observed grain-size and millennialscale Atlantic events suggests that the loess records in Central Asia represent a response not only to global signals but also local signals, such as glacial activity and local weathering. This overlap will weaken the global signals.

**C&S**: L 483-484: This sentence is not clear: is 'which' referring to the conclusions by the authors or by Vandenberghe et al.? It is not clear therefore what really is contradicting

**A&R**: I am so sorry for the poor expression. The 'which' referred to the conclusions by the authors. We have revised the sentence, like this "Darai Kalon is located in a region where the mid-latitude westerlies clearly have a much stronger influence, especially during full glacial conditions (Vandenberghe et al., 2006). In contrast, our results from the Ili Basin suggest that the mid-latitude westerlies did not always predominate north of the Kyrgyz Tian Shan due to northward or southward movement of the climate subsystem. In this case, the high mountains in Central Asia most likely obstructed the migration of the Asiatic polar front further south towards Tajikistan where those data were derived (Machalett et al., 2008), thereby resulting in a stronger westerlies signal at Darai Kalon than at NLK." The movement northward or southward of mid-latitude westerlies makes the Ili Basin more sensitive to paleoclimate change in Central Asia, which establishes the strategic position of the Ili Basin in paleoclimatic reconstruction.

However, as we mentioned above, the Siberian high-pressure systems predominate in the Ili Basin during cold phases, leading to dust transport and increased loess accumulation at NLK, and our grain-size proxy data can also correlate with abrupt events, such as Heinrich events (H1 to H6) identified from the North Atlantic records, though EM1 proportions fluctuate weaker in H2 and H5 events. Therefore, we have revised this section.

For *Technical comments*

**C&S**: L 123: 'more reliable' than what?

**A&R**: Thank you for your careful reading. We have rewritten this sentence. Actually, we mean that the optically stimulated luminescence (OSL) dating is more reliable for constructing a loess chronology than AMS $^{14}$C ages for older than MIS2 aeolian sediments according to Song et al. (2015).

**C&S**: L 317: 'shorter' than what?

**A&R**: EM1 is likely derived from shorter distance transport of suspended load owing to its larger modal grain size. Thus, its transport distance is shorter than the finer grains, like the EM2 and EM3 fractions in this manuscript.

**C&S**: L 139: insert 'were' between 'S1)' and 'then'; Figure 1 is too small.
   L 182: remove 'are';
   L 513: remove 'can' or 'may'.

**A&R**: Yes, these are grammar errors. We have corrected these mistakes accordingly. We also adjusted the layout of Fig. 1 and increased front size. Thank you.

**C&S**: I suggest to shorten the title a bit

**A&R**: Yes, we have rewritten the title, "Aeolian dust dispersal patterns since the last glacial period in eastern Central Asia: Insights from a loess-paleosol sequence in the Ili Basin". Thank you.

**C&S**: Dear authors, I agree with most of your replies and thank you for the modifications. I just want to react with 2 comments: 1. To the origin of the very fine silt-clay component: Chemical weathering is indeed a good candidate as measured by Konert and Vandenberghe 1997, and well-illustrated by the experiments of Sun YB et al 2006. Transport as aggregates of fines by monsoonal dust storms (Qiang et al 2010) is contradicted by their very widespread and general occurrence (Vandenberhe 2013). Adherence of fines to larger grains has been contradicted by several authors. 2. Provenance of EM 1-2: I agree with your explanation. I understand now that you also agree with a northern wind, however not crossing the high mountains to the north but carrying dust only at low elevation over short distance. In my opinion, the carrying agent may still be the northern monsoonal wind, although restricted to the Ili basin.

**A&R**:

1. The complexity of finer component is reflected in not only its origin but also uncertainty of instrument measurement (Ujvari et al., 2016;Mason et al., 2011). Chemical weathering can efficiently decrease gain size of loess (or paleosol) through the transformation of feldspar minerals into clay minerals linked closely to the process of pedogenesis. Sun et al. (2011) regarded this component formed by pedogenesis "ultrafine component". However, we have investigated clay mineralogy of NLK loess section, and the results show that the major clay mineral components in the NLK section were illite, chlorite, kaolinite and smectite, and that those clay minerals mainly had detrital origin, and rather than are in-situ weathered products. Moreover, variations in illite contents along the NLK section may be controlled by wind intensity, because weaker wind intensity would transport more fine fractions, which was supported by the wind tunnel experiment (Wang et al., 2017). Therefore, we think the degree of influence of chemical weathering on the loess grain size depends on the differences of environment conditions from site to site. Qiang et al. (2010) suggested that formation of aggregation increased particle mass, which enabled fine grains to be deposited even under stronger winds by dry deposition, however, the aggregates had larger pores and relevant lower density than individual minerals grain of the same size. Therefore the aggregates still can be influenced by the effects of sorting by aeolian processes. However, by observing the dust deposition collected in dust storm, Lin et al. (2016) thought that particles less than 20 µm could settle down during floating dust weather when the wind velocity decreased and even stopped. Therefore, it seemed to be difficult to distinguish that the aggregates were formed after deposition or they were transported by winds directly. Observations of modern dust under the scanning electron showed the phenomena of aggregation and/or fine particles adhering to larger ones (Pye, 1995, 1987;Derbyshire et al., 1998;Falkovich et al., 2001;Qiang et al., 2010), whereas the micrographs of fresh samples from the southern margin of Tarim Basin under SEM showed little aggregation, or adhering of fine particles to the coarse particles (Lin et al., 2016). Maybe more convincing evidence will come from a lot of studies of modern storm processes.

2. Yes, we agree with you. After reconsideration as mentioned above, we suggest that the Siberian high-pressure system exerts a significant influence on wind dynamics and thus the loess deposition in the eastern Ili Basin. Therefore, we have rewritten the section 5.3. Please see *Lines 439-487* in the revised manuscript.

***Replies to the comments of anonymous Referee #2***

For *Linguistic issues*

**C&S:** Lines 37-42: A lack of correlation between EM1 proportions and GISP $\delta^{18}O$ values at the millennial scale, combined with modern weather data, suggests that Arctic polar front predominates in the Ili Basin and the Kyrgyz Tian Shan piedmont during cold phases, which leads to the dust transport and accumulation of loess deposits, while the shift of mid-latitude westerlies towards the south and north controls the patterns of precipitation/moisture variations in this region. Reviewer's note: a lack of correlation between A and B means C was dominant? It implies that there are no other possibilities (D, E, …). Even worse, is "while the shift of the mid-latitude westerlies … controls patterns of precipitation/moisture …" corresponding to or with "shift of the Arctic polar front controls the temperature patterns of wind strength"? If so, you have to say so.

**A&R:** Thanks for your critical comments. The logic of the abstract was unclear, and it is unreasonable to draw conclusions beyond the information available in the data. We have now tried to clear the confused logic, and rewritten the Abstract and Conclusions sections in this manuscript. It is important to note that Central Asia is very large and consequently it is reasonable to assume that different climate subsystems act upon different parts of the region. Therefore, observations made at one end of Central Asia (e.g. Tajikistan) do not necessarily apply to the other (e.g. Ili Basin). Furthermore, the Ili Basin itself is almost 1000 km across and is geographically diverse, and it is reasonable to assume that the western part of the basin, e.g. the published site of Remizovka, is more exposed to influences such as the polar northerlies than sites in the eastern part of the basin, e.g. NLK presented here, which are much more sheltered by the high Tien Shan mountains.

Tajikistan is mainly impacted by the westerlies, and the North Atlantic climatic signals are presented in Tajikistan loess, which implies that the westerlies linking the North Atlantic and the Eurasia loess, can influence accumulation of loess deposits in Tajikistan (Vandenberghe et al., 2006). A lack of good correlation between EM1 proportions and GISP $\delta^{18}O$ values at the millennial scale only indicates that other climate systems control the wind dynamics responsible for dust transport and the accumulation of loess during cold phases in NLK, rather than the Westerlies. Thus, we cannot conclude that "Arctic polar front predominates in the Ili Basin and the Kyrgyz Tian Shan piedmont during cold phases."

We added some records from modern and Holocene climate change records to substantiate our arguments for mid-Westerlies changes. Actually, those records demonstrated that the mid-latitude Westerlies truly controlled the patterns of moisture variations in Arid Central Asia (ACA) (Huang et al., 2015;Li et al., 2011b;Cai et al., 2017). However, we can't draw that conclusion from "A lack of correlation between EM1 proportions and GISP $\delta^{18}O$ values at the millennial scale".

"while the shift of the mid-latitude westerlies … controls patterns of precipitation/moisture …" isn't corresponding to or with "shift of the Arctic polar front controls the temperature patterns of wind strength". Available data enable us to compare our data from the eastern, sheltered end of the Ili Basin with the more exposed Remizovka section at the southwestern margins of the basin – with respect to likely climatic influence and its impact on grain size. Remisowka (Machalett et al., 2008) is located along the northern piedmont of Tianshan Mountains (Fig. 1a in manuscript). Because NLK is much more sheltered from northerly weather systems than Remisowka, there is a good chance that the polar front had more of an influence on Remisowka than on NLK. Furthermore, based on modern and Holocene climate data and comparison of the EM1 proportions and mean grain size (MGS) data from the Jingyuan section in northwestern CLP, we argue that the Siberian High may have exerted a significant influence on wind dynamics in the Ili Basin, leading to dust transport and the accumulation of loess during cold phases in NLK. Therefore, we argue that the Siberian High controls wind strength and mid-latitude westerlies control precipitation/moisture. A strengthened Siberian High would push the mid-latitude Westerlies pathways further to the south, resulting in comparably drier conditions in the northern Central Asia regions (e.g. Tianshan Mountains) but wetter conditions in south-western Central Asia (Pamir) (Lei et al., 2014;Wolff et al., 2017). Intensity and geographical position of the Siberian High can strongly control precipitation and atmospheric circulation patterns (meridional or zonal) at mid-latitudes of Asia (Panagiotopoulos et al., 2005). The coupling of the Siberian High with the mid-latitude Westerlies system likely contributed significantly to the climate variability in the study area. We have modified our text to explain these drivers more clearly, and also rewritten the *Abstract*. Please see details in the revised manuscript

**C&S**: Lines 42-44: Comparison of EM1 proportions with Northern Hemisphere summer insolation clearly illustrates local insolation-based control on wind dynamics in the region, and humidity can also influence grain size of loess over MIS3 in particular. Reviewer's note: to me (this reviewer), the logic relationship between these two sentences are not traceable at all. "local insolation-based control on wind dynamics": what does this mean?

**A&R**: We reconsider the relationships between June insolation at 45ºN and EM1 proportions in Fig. 7, and the reasons for variations in EM1 proportions. We think it is more reasonable to consider that the availabilities of source sediments which are more likely impacted by development of permafrost and vegetation growth in dust source areas, are responsible for the weaker EM1 proportion fluctuations in LGM and early-MIS3. Therefore, we have deleted the discussion of summer insolation. Please see the details in line 464-470 and 504-509 of revised manuscript.

**C&S**: Lines 55-60: The relative influence and intensity of these major climate subsystems have varied across the latitudinal and longitudinal range of Central Asia through time. Thus identification of the predominant climate regimes in a certain region is a crucial precondition for tracing paleoclimatic evolution. Reviewer's note: (1) relative influence? Maybe relative importance. (2) The first sentence continues its SPECIFIC tone (i.e., Central Asian), but the second sentence turns to a general tone (i.e., a certain region). To me (this reviewer), it is misleading.

**A&R**: (1) We have clarified this distinction in the text, and substituted "relative influence" with "relative importance". (2) We cannot use a specific concept to represent a general concept. It is indeed misleading. We have changed the second sentence to "Thus identification of the predominant climate regimes in this region, using geological archives, is a crucial precondition for tracing paleoclimatic evolution."

**C&S**: Lines 66-72: While loess in Central Asia has (……) increasingly formed the focus of loess research, as yet the forcing mechanisms and the climatic conditions responsible for loess-paleosol sequences formation are ambiguous, and the paleoclimatic evolution recorded by these loess deposits in this region is not systematically understood. Reviewer's note: to me (this reviewer), "increasingly formed the focus", "the forcing mechanisms … are ambiguous", and "not systematically understood" are all belong to"expression inadequacies".

**A&R**: Here, we have simplified the language and made the purpose of this paper much clearer and better to understand. We also added three citations in an effort to reinforce the lack of systematic understanding of the forcing mechanisms and the climatic conditions responsible for loess-paleosol sequences formation.

**C&S**: Lines 78-81: Climatic teleconnections, especially between the North Atlantic and East Asian Monsoon regions, are likely to have been recorded within the Central Asian loess. As yet, however, the region so far largely lacks data by which the role and contribution of the central parts of the Eurasian continent, as an environmental bridge, can be elucidated. Reviewer's note: to me (this reviewer), there is a logic gap in this statement. I mean that you (authors) may have to bring the environmental bridge to the front so that the importance of Central Asia in documenting the teleconnections is pronounced first.

**A&R**: Thanks for your suggestion. We have clarified the language in the text and the wording of our arguments. Since we know basically nothing about millennial-scale climatic changes in Central Asia, our aim is to investigate a loess section in Central Asia to see to what degree climatic teleconnections exist between North Atlantic and East Asia first, i.e. the first step is to generate data. Therefore, we have made the aim clearer, like this "Data for Central Asian loess are so far lacking at this resolution, despite its strategic location as a likely environmental bridge between the North Atlantic and East Asian Monsoon regions." We deleted the sentence "Climatic teleconnections, especially between the North Atlantic and East Asian Monsoon regions, are likely to have been recorded within the Central Asian loess."

**C&S**: Other suggestions Magnetic Susceptibility 1.1. "Low susceptibility in paleosols and high susceptibility in loess units" were sufficiently documented in Alaskan loess and in Siberian loess and Professor Liu Xiuming is a leading scientist on this. Please see if his works and propositions can help you. 1.2. The coarse particle-association of high susceptibility can be tested simply by measuring the susceptibility of different particle size fractions. This can be done on selected samples and the data of the selected samples may elevate your confidence of interpretation. 1.3. If I were the author, I would have completely excluded susceptibility portion from this paper and may (just may) write a separate paper on magnetic susceptibility.

**A&R**: The relationship between pedogenesis and magnetic susceptibility in the higher-latitude loess deposits of Alaska and Siberia is different from the Chinese Loess Plateau loess as suggested by Liu et al. (1999) and Liu et al. (2008). At NLK, lower susceptibility exists in paleosols and higher susceptibility in loess units. Although this scenario is difficult to explain fully through variation in wind strength alone, it showed that wind strength, or wind dynamics, would influence MS variations at least and thus paleoclimatic reconstruction using climatic proxies, such as MS. Thus it is necessary to understand the atmospheric dynamic pattern during loess deposition further.

Consistently low $\chi_{fd}$% values in both loess and paleosol layers demonstrated that the content of SP particles is very low, and consequently that their contribution to MS can be ignored. That is, weaker pedogenesis prevents the efficient production of SP grains. We hence consider that allogenetic magnetic minerals made the greater contributions to MS, and these correlate with dust transportation. Following the reviewer's suggestions, we sieved five samples into > 63 µm, 63 – 40 µm, 40 – 32 µm, 32 – 20 µm and < 20 µm grain size fractions, and each of samples were measured at least three times using a Bartington MS2 meter. Our results showed lower MS values in > 63 µm grain size fractions and maximum values existed in 32 – 20 µm or < 20 µm grain size fractions (Table R1), which indicated that the major ferromagnetic minerals were always smaller than sand size. Therefore, we deleted the sentence "MS enhancement at NLK is primarily driven by increased concentrations of sand-sized detrital magnetic minerals" in the manuscript.

Understanding the mechanisms for the enhancement of magnetic susceptibility is beyond the scope of this study. We only intended to illustrate the significant impacts of wind dynamic on MS. In addition, ferromagnetic minerals, including magnetite and hematite, belong to heavy minerals which have higher relative density. Thus when wind becomes stronger, more ferromagnetic minerals will be transported to deposition areas, resulting to higher MS values. Thus we have modified the subtitle 5.1, like this "Impacts of wind strength on magnetic susceptibility variations".

Table R1 Magnetic susceptibility values ($10^{-8}m^3kg^{-1}$) of different grain-size fractions

| Sample No. | Grain-size Fractions | | | | |
|---|---|---|---|---|---|
| | > 63 µm | 63-40 µm | 40-32 µm | 32-20 µm | <20 µm |
| NLK400 | 72 | 66 | 86 | 88 | **98** |
| NLK800 | 67 | 61 | 80 | **87** | 85 |
| NLK1108 | 73 | 78 | 104 | 107 | **108** |
| NLK1110 | 75 | 68 | 96 | **109** | 105 |
| NLK1400 | 79 | 83 | 94 | 100 | **111** |

Note: Red font represents maximum values in different grain-size fractions.

**C&S**: Particle Size 2.1. You need a comprehensive and streamlined review on existing literature dealing with interpretation of loess particle size. The literature review can be either "school division-based" or time-based (earlier time and later time) or country based (west and China). 2.2. After the expected review is properly done, you may delete those insignificant references (I mean that you cited too many and that many of them may be insignificant). 2.3. Since you heavily rely on Vandenburghe (2013) for EM1, EM2, and EM3 arguments, you are strongly suggested to provide a complete and concise re-statement of Vandenburghe (2013) in debating pros and cons of EM1, EM2, and EM3 for representing aolian dynamics. If he was so sure and nobody else was at his odd, your application of EM1, EM2, and EM3 to interpreting aolian dynamics may be more acceptable. If his argument was case-dependent, you have a harder task to establish your case though. 2.4. I am wondering if the cumulative particle-size curve does show a statistically meaningful break between EM1 and EM2 and also a break between EM3 and EM2? If it does not, should your reliance on

Vandenburghe (2013) be questionable? What I try to say is: if you can confidently justify the acceptance of EM1, EM2, and EM3 for representing aolian dynamics, you do have a case here. Otherwise, your opponents can always argue that: those coarse particles may have indeed locally sourced, but those fine particles can either be remotely (high-elevation) sourced or locally (near-surface) sourced.

**A&R**: Thanks for your suggestions. In the section 5.2, we have summarized the significance of grain-size analysis. Relevant studies were separated into two groups according to the unmixing method of grain size spectra. Vandenberghe (2013) applied visual inspection of grain-size distribution curves and the EMMA end-member analysis in combination to define the characteristic grain-size distribution of primary loess deposits and review their respective processes and conditions of transports and deposition, relying largely on loess samples from central and eastern Asia and northwestern and central Europe. Thus his argument was based on a large number of previous studies from a range of sites, and is not case-dependent. For example, *the subgroup 1.b.2* in Vandenberghe (2013) has also been identified in the loess of Chinese Loess Plateau, southern, northwestern and central Europe. Furthermore, in the studies of loess sediments from the Qilian Mountain region, Rasmussen et al. (2014), Nottebaum et al. (2015) and Yang et al. (2016) have interpreted the multiple sources of loess sediments and dynamic conditions according to sediment groups in Vandenberghe (2013). We have included those arguments in the main text.

EM1 and EM2 of our results have modal grain size approximately corresponding to the '*subgroup 1.b.1*' and '*subgroup 1.b.2*' respectively. Vandenberghe (2013) suggested that although component 1.b.1 and 1.b.2 occur jointly together in the proximal depositional regions, they are clearly distinct from each other in terms of the coverage and transportation distance. In Fig. 4 of manuscript, the mirror image relationships over millennial scales can be observed, which may implied that both EM1 and EM2 have a same origin, but wind strength controlled the relative proportions of both through time. In addition, grain-size distributions of modern dust illustrate a modal grain size of 33.3 µm in winter and 44.6 µm in summer in the northern and western Chinese Loess Plateau (Sun et al., 2003) (Fig. R3). These modes are similar to EM2 and EM1 in our results, respectively. It is generally assumed that vegetation coverage is more extensive in summer than in winter in CLP. Therefore, availability of sediments in source areas wouldn't influence the grain sizes, conversely differences in wind dynamic between these two seasons likely play an important role in controlling the grain sizes. While EM3 ("*subgroup 1.c.1*") indicated a different aerodynamic environment from EM2. The former would settle when the wind velocity decreases and even stops, as suggested by Lin et al. (2016), but the latter were interpreted as transportation during cyclonal dust storm outbreaks (Vandenberghe, 2013). Consequently, the cumulative particle-size curve can give a statistically meaningful break between EM1 and EM2 and also a break between EM3 and EM2.

[Figure]

Fig. R3 Grain-size distributions of seasonal dusts in the northern and western CLP (Huanxian)
Actually, greater dispute exists in the origin of the EM3 size fraction. In the manuscript, we suggest that the fine-grained EM3 (c. 18.9 µm) is the result of background loess supply in the Ili Basin, and infer the EM3 modal peak to derive from low altitude non-dust storm processes after excluding the aggregate model, transportation by high-altitude westerlies and influences of post-depositional processes. Therefore, those fine particles are also likely to be locally (near-surface) sourced.

**C&S**: Questions for 5.3 Aeolian dust dynamics in eastern Central Asia: links to atmospheric systems Lines 440-447: Central Asia is variably influenced by the Asian monsoon from the south (Dettman et al., 2001; Cheng et al., 2012), the mid-latitude westerlies (Vandenberghe et al., 2006), the Siberian high-pressure systems from the northeast (Youn et al., 2014), and the polar front from the north (Machalett et al., 2008). However, by virtue of its geographical position, most of these climate influences can be excluded for the Ili Valley since it is sheltered to the northeast, east and south. The Asian high mountains largely inhibit the intrusion of Asian (Indian and East Asian) monsoons to the region, and the influence of the Siberian High (An, 2000) has been shown to decrease westward from the CLP (Vandenberghe et al., 2006). Reviewer's note: Downplaying Asian monsoons may be acceptable since the Yili Valley is indeed blocked by the Tianshan Mountains on the south. But, downplaying Siberian high-pressure system (SibH) is not well justified. Yes, SibH is weakening away from its center, but you cannot say that the Yili Valley was beyond the SibH influence. Furthermore, your favored "polar front" is actually also blocked by high mountains on the north. If polar front was indeed the major player, you may have to provide modern climate backgrounds in which strong polar front interacted with the prevailing westerly flow to stimulate dust storms in the Yili Valley.

**A&R:** Thanks for your good suggestions. As described above, the Siberian high-pressure systems predominate in the Ili Basin during cold phases, which leads to dust transport and increased loess accumulation at NLK, and the mid-latitude Westerlies controlled broad-scale patterns of moisture variation across ACA.
Modern meteorological data show that the maximum wind at NLK mainly blows from the west, and that dust storm development in Ili river valley is closely linked with southward-moving high-latitude air masses, while the air masses can enter into the Ili Basin round the northern Tianshan (see the Supplementary materials and Ye et al. (2003)). Therefore, the Siberian high-pressure system is able to influence the Ili Basin, and the southward-moving high-latitude air masses associated with it can enter into the Ili Basin, leading to dust transport and the accumulation of loess deposits during cold phases in NLK. The coupling of the Siberian High with the mid-latitude Westerlies system likely contributed significantly to the climate variability at NLK in the eastern Ili Basin.

We have rewritten the section 5.3. Please see the details in the revised manuscript and Supplementary materials.

**C&S**: Lines 448-456: Modern satellite data indicates that dust storm development in Ili river valley is closely linked with southward-moving high-latitude air masses (Ye et al., 2003). Karger et al. (2016) provided a detailed picture of the westerlies for the Ili Basin, in which a rain belt gradually migrated towards the south and north in autumn and summer, respectively. According to this scenario, enhanced evaporation coupled with strengthened westerly winds would bring more humid and warm air masses to Arid Central Asia (ACA) during the Holocene (Zhang et al., 2016). Therefore, based on our grain-size observations, we argue that the Arctic polar front, intruding southward in the winter and retracting northward in summer (Machalett et al., 2008), most likely increased the frequency and strength of cyclonic storms, leading to dust transport and the accumulation of loess deposits during cold phases when it predominated in the Ili Basin and along the Kyrgyz Tian Shan piedmont. Reviewer's note: I (this reviewer) failed to see the linkage between "souward-moving high-latitude air masses" and "migrated rain belt". I also failed to see the linkage between "enhanced evaporation" and "strengthened westerly winds". Consequently, I failed to see the logic of your reasoning: the Arctic polar front, intruding southward in the winter and retracting northward in summer (Machalett et al., 2008), most likely increased the frequency and strength of cyclonic storms during cold phases. At least, you have to say more about the logic of your reasoning.

**A&R:** In this respect our logic was flawed. We have clarified the logic of our arguments in the text. As mentioned above, it is unreasonable to draw conclusions beyond the information available in the data. Therefore, we reconsidered the atmospheric system responsible for aeolian dust dynamics in our study area, and then rewrote and rearranged the paragraphs.

As explained above, the Siberian high-pressure systems exerted a significant influence on wind dynamics responsible for dust transport and the accumulation of loess deposits during cold phases in NLK, and the mid-latitude Westerlies controlled the patterns of moisture variations in Arid Central Asia (ACA), based on modern and Holocene climate data and comparison of the EM1 proportions and mean grain size (MGS) data from the Jingyuan section in northwestern CLP. A strengthened Siberian High would push the mid-latitude Westerlies pathways further to the south, which resulted in comparably drier conditions in the northern Central Asia regions (e.g. Tianshan Mountains) but wetter conditions in south-western Central Asia (Pamir) (Lei et al., 2014;Wolff et al., 2017). Intensity and geographical position of the Siberian High can strongly control precipitation and atmospheric circulation patterns (meridional or zonal) at mid-latitudes of Asia (Panagiotopoulos et al., 2005). Therefore, the coupling of the Siberian High with the mid-latitude Westerlies system likely contributed significantly to the climate variability in the study area, which may interpret the seesaw relationship during MIS3 shown in Fig. 7 of manuscript.

*Replies to the comments of editor*

**C&S**: Please revise your manuscript by incorporating the changes that you have made in response to the referees' comments. My concern with this manuscript is that the discussions and conclusions are based on one loess section. The title with the words such as environmental dynamics doesn't really reflect the level of science that you try to convey in the manuscript. Please re-consider the title.

**A&R**: We have rechecked and further considered the correlation of EM1 proportions and GISP $\delta^{18}$O from the Greenland ice cores. Although it may not be possible to reliably match fluctuations in loess records to millennial climatic events due to the limitations in dating techniques in loess research, our grain-size proxy data can still correlate with abrupt events, such as Heinrich events (H1 to H6) identified from the North Atlantic records (Fig. 7 in the revised manuscript). However, EM1 proportions fluctuate weaker in H2 and H5 events, which we attribute to availabilities of source sediments, as mentioned above. These millennial-scale events were also found in Xiaoerbulake section, Talede section and Zhaosu section from the Ili Basin (Li et al., 2016a;Li et al., 2011a;Zhang et al., 2015). However, in our opinion, the Siberian high-pressure systems predominate in the Ili Basin during cold phases, which leads to dust transport and increased loess accumulation at NLK and is responsible for those North Atlantic millennial scale abrupt climate events. Therefore, our data support the Siberian High can also transport the climatic signals in the North Atlantic to the East Asia, via the ice sheets in high northern latitudes. Moreover, lack of good correlation between EM1 proportions and GISP $\delta^{18}$O values during relatively mild interstadial periods (Dansgaard-Oeschger cycles) when the mid-latitude westerlies shift northwards, implies the minor direct influences of the mid-latitude westerlies on the loess accumulation in NLK, which is not in agreement with the previous studies.

In addition, some of the peaks in EM1 curve correspond to valleys in GISP $\delta^{18}$O curve (black arrows in Fig. 7 in the manuscript) except Heinrich events, yet many do not (pink dashed lines in Fig. 7 in the manuscript). The same case also occurs in the Western CLP (Chen et al., 1997), that is, all of the Heinrich events occurred during periods of strong winter monsoon in China, but not all of the periods of strong winter monsoon in China correlate with Heinrich events in the North Atlantic. The differences may be because the loess records in our study area represent a response not only to global signals but also local signals such as local atmospheric circulation and topography.

We have rewritten the title, "Aeolian dust dispersal patterns since the last glacial period in eastern Central Asia: Insights from a loess-paleosol sequence in the Ili Basin, northwest China". Thank you.

[revised manuscript text omitted]

Fig02

[Figure]

Fig03

[Figure]

Fig04

[Figure]

[Figure]

Fig05

[Figure]

Fig06

[Figure]

Fig07

[Figure]